

# Stratospheric ozone loss in the Arctic winters between 2005 and 2013 derived with ACE-FTS measurements

Debora Griffin[1], Kaley A. Walker[1, 2], Ingo Wohltmann[3], Sandip S. Dhomse[4,5], Markus Rex[3], Martyn P. Chipperfield[4,5], Wuhu Feng[4,6], Gloria L. Manney[7, 8], Jane Liu[9,10], and David Tarasick[11]

[1]Department of Physics, University of Toronto, Toronto, Ontario, M5S 1A7, Canada
[2]Department of Chemistry, University of Waterloo, Waterloo, Ontario, N2L 3G1, Canada
[3]Alfred Wegener Institute for Polar and Marine Research, 14401 Potsdam, Germany
[4]Institute for Climate and Atmospheric Science, University of Leeds, Leeds, L2S 9JT, UK
[5]National Centre for Earth Observation, University of Leeds, Leeds, L2S 9JT, UK
[6]National Centre for Atmospheric Science, University of Leeds, Leeds, L2S 9JT, UK
[7]NorthWest Research Associates, Socorro, New Mexico, USA
[8]Department of Physics, New Mexico Institute of Mining and Technology, Socorro, New Mexico 87801, USA
[9]Department of Geography and Program in Planning, University of Toronto, Toronto, Ontario, M5S 3G3, Canada
[10]Nanjing University, Nanjing, Jiangsu, 210023, China
[11]Science and Technology Branch, Environment and Climate Change Canada, Toronto, Ontario, M3H 5T3, Canada

*Correspondence to:* Kaley A. Walker
(kaley.walker@utoronto.ca)



**Abstract.** Stratospheric ozone loss inside the Arctic polar vortex for the winters between 2004/2005 and 2012/2013 has been quantified using measurements from the space-borne Atmospheric Chemistry Experiment Fourier Transform Spectrometer (ACE-FTS). Six different methods, including tracer-tracer correlation, artificial tracer correlation, average vortex profile descent, and passive subtraction with model output from both Lagrangian and Eulerian chemical transport models (CTMs), have been employed to determine the Arctic ozone loss (mixing ratio loss profiles and the partial column ozone losses between 380 K and 550 K). For the tracer-tracer, the artificial tracer, and the average vortex profile descent approaches, various tracers have been used. Here, we show that $CH_4$, $N_2O$, HF, and CFC-12 are suitable tracers for investigating polar stratospheric ozone depletion with ACE-FTS. The ozone loss estimates (in terms of the mixing ratio as well as total column ozone) are generally in good agreement between the different methods and among the different tracers. However, the tracer-tracer correlation method does not agree with the other estimation methods in March 2005 and using the average vortex profile descent technique typically leads to smaller maximum losses compared to all other methods. The passive subtraction method using output from CTMs generally results in smaller uncertainties and slightly larger losses compared to the techniques that use ACE-FTS measurements only. The ozone loss computed, using both measurements and models, shows the greatest loss during the 2010/2011 Arctic winter. For that year, our results show that maximum ozone loss (2.1-2.7 ppmv) occurred at 460 K. The estimated partial column ozone loss inside the polar vortex (between 380 K and 550 K) is 66-103 DU, 61-95 DU, 59-96 DU, 41-89 DU, and 85-122 DU for March 2005, 2007, 2008, 2010, and 2011, respectively. Ozone loss is difficult to diagnose during 2005/2006, 2008/2009, 2011/2012, and 2012/2013 because strong polar vortex disturbance or major sudden stratospheric warming events significantly perturbed the polar vortex thereby limiting the number of measurements available for the analysis.

## 1 Introduction

Arctic ozone column loss is extremely variable and can range from near zero to about 150 DU (e.g., Manney et al., 2011; Kuttippurath et al., 2012; Livesey et al., 2015), unlike in the Antarctic, where ozone loss is typically large, and shows smaller interannual variability (e.g., WMO, 2014). The large interannual variability is mostly caused by the Arctic dynamics and meteorology (e.g., Andrews, 1989; Schoeberl and Hartmann, 1991; Schoeberl et al., 1992). Due to topography and land-sea contrasts, wintertime wave activity that drives stratospheric circulation is much stronger and more variable in the Northern Hemisphere (NH) than in the Southern Hemisphere (SH) (e.g., Weber et al., 2003). Therefore, the polar vortex in the NH is typically weaker and more variable from year-to-year than the polar vortex of the SH. Climatologically, the Arctic lower stratospheric polar vortex forms in November and breaks up in April, but break up dates can be much earlier (when there are major sudden stratospheric warmings (SSWs), during which temperatures increase rapidly and mid-stratospheric zonal mean winds reverse) or later (in particularly quiescent winters) (WMO, 2014). If, however, the polar vortex remains stable and temperatures within it low, polar stratospheric clouds (PSCs) can form (e.g., Steele et al., 1983; Toon et al., 1986; Crutzen and Arnold, 1986; Lowe and MacKenzie, 2008). PSCs that contain primarily ice particles (Steele et al., 1983) typically form at temperatures below 188 K (Poole and McCormick, 1988). Other PSCs are composed of either ice and nitric acid trihydrate (NAT) particles, or super-cooled ternary solution (STS), a mixture of $HNO_3$-$H_2SO_4$-$H_2O$ particles, and can form at much





higher temperatures around 195-197 K (e.g., Crutzen and Arnold, 1986; Toon et al., 1986; Arnold, 1992; Pitts et al., 2007, 2009, 2013). Since wintertime temperatures in the Arctic polar regions are higher than those in the Antarctic winter, most PSCs in the Arctic are nitric-acid containing ones that form at higher temperatures. (Solomon, 1999, and references therein). Chlorine activation is triggered on the surface of PSCs and/or cold binary aerosols (e.g., Portmann et al., 1996; Drdla and Müller, 2012;

WMO, 2014), releasing chlorine molecules. When the chlorine molecules are exposed to sunlight, these molecules break into chlorine radicals that are responsible for springtime polar catalytic ozone depletion (e.g., McElroy et al., 1986; Solomon et al., 1986; Molina and Molina, 1987). Thus, the amount of yearly ozone loss in the Arctic is strongly influenced by the temperature within the polar vortex and whether an SSW event occurred.

In recent years, there have been several major SSWs in the Arctic: the most pronounced SSW events occurred in January

2006 (Manney et al., 2008a; Coy et al, 2009; Manney et al., 2009a), January 2009 (Labitzke and Kunze, 2009; Manney et al., 2009b), in January/February 2010 and January 2013 (Manney et al., 2015; Coy and Pawson, 2015). In those years, the polar vortex broke up in January and no significant springtime chemical ozone depletion was detected. In January 2012, very strong polar vortex disturbance occurred, likely due to a Arctic Polar-Night Jet Oscillation Event (Berhard et al., 2012; Chandran et al., 2013; Hitchcock et al., 2013). During the 2010 winter, the polar vortex was highly influenced by dynamics and mixing due

to a major SSW: the vortex split in two parts in mid-December 2009, these two parts reunited in January, and in mid-February the vortex split again into two parts and reunited at the beginning of March (e.g., Dörnbrack et al., 2012; Kuttippurath and Nikulin, 2012; Wohltmann et al., 2013). In the Arctic spring if a polar vortex exists, the ozone mixing ratio inside that polar vortex peaks at around 3.5 ppmv between approximately 450 K and 475 K in the absence of chemical ozone depletion (based on ACE-FTS measurements inside the polar vortex in January). During the winters of 2004/2005 (Manney et al., 2006; Jin et al.,

2006; Kuttippurath et al., 2010), 2006/2007 (Kuttippurath et al., 2010), and 2007/2008 (Kuttippurath et al., 2010), the Arctic polar vortex was strong and ozone depletion on the order of 1.5 ppmv (around 40 % loss) occurred in the lower stratosphere. In the winter of 2010/2011, a very strong vortex and exceptionally prolonged cold period led to unprecedented Arctic chemical ozone loss (Balis et al., 2011; Manney et al., 2011; Sinnhuber et al., 2011; Adams et al., 2012; Arnone et al., 2012; Kuttippurath et al., 2012; Lindenmaier et al., 2012). The chemical ozone loss peaked by the end of March at around 2.5 ppmv (around 70 %

loss) in the lower stratosphere.

In addition to chemical ozone depletion, dynamical processes, such as descent, mixing of extra-vortex air, and mixing within the polar vortex affect the ozone concentration. Because of the dynamical variability of the Arctic polar vortex, quantifying chemical ozone loss in the Arctic is challenging. As such, the effects of chemical loss versus dynamics need to be understood and separated (e.g., Manney et al., 1994a, 1995; Chipperfield and Jones, 1999; Harris et al., 2002; WMO, 2006; Livesey et

al., 2015). To investigate the change in ozone due to chemical processes, various methods can be used. Several approaches have been developed to estimate the springtime ozone abundance profile that results solely from dynamical processes. The difference between this estimated "passive ozone", which is only influenced by dynamics (and not by chemical processes), and the observed ozone is assumed to be the chemical ozone loss.

Some methods, such as the tracer-tracer correlation approach (e.g., Proffitt et al., 1993; Müller et al., 2001), only require

measurements to determine the passive ozone. The tracer-tracer correlation method determines the chemical depletion from the





relationship between ozone and a long-lived passive tracer. However, processes that mix extra-vortex air into the polar vortex, as well as descent from higher altitudes are not considered in this approach, and these can change the tracer-tracer correlation significantly, and thus render the tracer-tracer correlation method inaccurate (e.g., Michelsen et al., 1998a, b; Plumb et al., 2000, 2003; Plumb, 2007). Using an artificial tracer (e.g., Esler and Waugh, 2002; Jin et al., 2006) that is constructed to be linearly

correlated with ozone can improve the accuracy of the loss estimate, since that linear relationship will not be changed by mixing processes (see Esler and Waugh (2002) for more details). Estimates can also be made by determining the average descent rate inside the polar vortex, obtained from a long-lived tracer, and then estimating the passive ozone abundance (e.g., Manney et al., 2006; Jin et al., 2006). This method can be applied in most years since descent is typically the dominant dynamical process in the Arctic vortex. Other methods use CTMs to determine the passive ozone profiles, where the ozone chemistry processes are

not included in the model run (e.g., Manney et al., 2005; Kuttippurath et al., 2010, 2012; Brakebusch et al., 2013; Wohltmann et al., 2013). The ozone loss can then be estimated from the difference between the modelled passive ozone and the observed (or modelled) ozone. These models also include ozone chemistry, and this output can be used to understand the accuracy of the simulations by comparing with observations. If no ozone chemistry is used for the model simulation, the difference between the measurement and the model can be estimated by comparing the passive ozone and the measurements at a time when no

significant ozone depletion is apparent.

The focus of this study is to use measurements from the Atmospheric Chemistry Experiment Fourier Transform Spectrometer (ACE-FTS; Bernath et al., 2005) between 2005 and 2013 to compare ozone loss estimates from different methods. Chemical ozone depletion for each spring is estimated using the tracer-tracer correlation method, the artificial tracer approach, the average vortex profile descent technique, the modelled passive ozone subtraction method using a Lagrangian and an Eulerian transport

model, and the passive subtraction method using only modelled ozone. Since ACE-FTS provides measurements of many trace gases, several tracers are investigated for the tracer correlation and descent approaches. This is the first study to evaluate these different ozone loss estimation methods based on a single observational dataset. Thus, the purpose of this work is to assess the differences in chemical ozone depletion obtained by different methods without the confounding influence of different trace gas datasets.

This paper is structured as follows: The ACE-FTS instrument and dataset are reviewed in Sect. 2, followed by a description of the methods used to estimate the springtime chemical ozone loss in Sect. 3. The results of the evaluation of the choice of tracer and the different methods are provided in Sect. 4. A comparison of results from this study with previous studies of Arctic ozone loss in spring 2011 is also given in Sect. 4. This is followed by a summary and conclusions in Sect. 5.

## 2 ACE-FTS measurements

### 2.1 ACE-FTS instrument and retrieval algorithm

The Atmospheric Chemistry Experiment (ACE), on SCISAT, was launched on 12 August 2003 and measurements have been taken since February 2004. The primary instrument on board SCISAT is the ACE-FTS, which measures the spectral region between 750 and $4400\,\mathrm{cm^{-1}}$ at a spectral resolution of $0.02\,\mathrm{cm^{-1}}$. The primary scientific objective of SCISAT is to improve



the understanding of polar ozone chemistry (Bernath et al., 2005). Therefore, the orbit of SCISAT was selected such that it provides measurements over the Arctic during the winter and springtime every year. The observation technique used by ACE-FTS is solar occultation, which provides profiles with a vertical resolution between 1.5 km and 6 km depending on the beta angle, the angle between the vector from the Earth to the Sun and the satellite velocity vector. Retrievals from the infrared

spectra provide profiles for over 30 atmospheric trace gases as well as the meteorological variables of temperature and pressure (Boone et al., 2005). The volume mixing ratio (VMR) of the various trace gas, temperature and pressure profiles used in this study are from the latest retrieval version, ACE-FTS v3.5 (Boone et al., 2013). The uncertainties provided with this dataset for the ACE-FTS profiles are statistical fitting errors from the retrieval algorithm. Systematic errors are not included (Boone et al., 2005). Profiles are retrieved from the top of the clouds up to approximately 150 km. For clear sky conditions, the lower limit

of the retrieved profiles can be as low as 5 km.

ACE-FTS ozone has been validated against various other space-borne as well as ground-based instruments. In the lower stratosphere (between approximately 14 km to 27 km, the region of interest for this study), generally good agreement with differences of less than $\pm 5\%$ was found between ACE-FTS v3.5 and the Aura Microwave Limb Sounder (MLS) and the Michelson Interferometer for Passive Atmospheric Sounding (MIPAS) ozone measurements (Sheese et al., 2016). The other

ACE-FTS trace gas retrievals that have been used in this study, such as $N_2O$, CFC-12 ($CCl_2F_2$), CFC-11 ($CCl_3F$), HF, $CH_4$, OCS, and CFC-113 have also been reported and validated in previous studies. Sheese et al. (2016) have shown that below 27 km differences between ACE-FTS v3.5 and MLS and MIPAS $N_2O$ measurements are within $\pm 10\%$. ACE-FTS $CCl_3F$ and $CCl_2F_2$ have been compared with MIPAS by Eckert et al. (2016), and these species agree to better than 15 % for $CCl_3F$ and 20 % for $CCl_2F_2$ in the altitude range of interest. HF has been compared to Halogen Occultation Experiment (HALOE)

observations and differences were within $10\%$ (Harrison et al., 2016). Some species have not been validated for the latest retrieval product. However, Waymark et al. (2013) have shown general improvements between the previous ACE-FTS v2.2 and the current ACE-FTS v3.0/3.5 across all baseline species. For the ACE-FTS v2.2+updates, the $CH_4$ mixing ratio is between $\pm 10\%$ of other space-borne instruments in the altitude range of interest here (De Mazière et al., 2008). OCS v2.2 has been compared with balloon-borne MkIV and shuttle-borne Atmospheric Trace Molecule Spectroscopy (ATMOS) measurements

in Barkley et al. (2008) and Velazco et al. (2011), and initial CFC-113 retrievals have been compared with ground-based measurements by Dufour et al. (2005).

### 2.2  Dataset used for the ozone loss estimates

The orbit of ACE-FTS, which was selected to observe the same latitudes in the same month every year, does not cover the whole globe at all times (Bernath et al., 2005). For example, measurements in the Arctic ($\geq 65°$N) are taken in approximately late

January, all of March, late May, mid July, mid September, and early October every year. For the ozone loss assessment in this study, ACE-FTS v3.5 measurements north of $65°$ between potential temperature 375 K and 550 K are considered. Quality flags, as recommended by Sheese et al. (2015), are used to remove physically unrealistic outliers and processing errors. Hereby, entire profiles have been removed from the dataset that contained quality flags between 4 and 7, as well as individual observations (within a profile) that contained a quality flag greater than 2. Version 1.1 of the ACE-FTS data quality flags was used.




Derived Meteorological Products (DMPs; Manney et al., 2007) are available at each 1-km tangent altitude within each ACE-FTS occultation. The geographical location can change significantly with tangent altitude for the ACE-FTS measurements. The geographical location of points from one ACE-FTS occultation, for altitudes between 15 and 25 km, can vary by up to $0.5°$ ($\sim 100\,\text{km}$) depending on the beta angle. The DMPs include information about the potential temperatures, as well as potential vorticity (PV), and are derived from GEOS version 5.2.0 analyses (GEOS-5; Rienecker et al., 2008).

In this study, ozone loss in March relative to January has been estimated inside the polar vortex. Thus, the ozone loss is estimated over a time period of approximately 1.5 months. Since some chemical ozone depletion can occur as early as December, most studies measure the chemical loss with respect to December. However, no December measurements are available at high latitudes from ACE-FTS, and therefore January was selected as the reference. The scaled potential vorticity (sPV; Dunkerton and Delisi, 1986; Manney et al., 1994b), from the DMPs is used to determine where the measurements were taken relative to the polar vortex. For March, measurements with $\text{sPV} \geq 1.6 \times 10^{-4}\,\text{s}^{-1}$ are selected as those located inside the polar vortex (Manney et al., 2007, 2008b). However, for January measurements, a more rigorous vortex selection criterion of $\text{sPV} \geq 1.8 \times 10^{-4}\,\text{s}^{-1}$ was applied. Since this criterion only considers measurements well inside the edge of the vortex, it reduces the influence of mixing from the vortex edge region and improves the results of the tracer-tracer method. Both sPV thresholds are toward the inside of the region of strong PV gradients demarking the vortex edge. These criteria have been applied to each method to be consistent throughout.

The time period investigated in this study is between 2005 and 2013. Ozone depletion could not be determined for all of those years. In 2004, no ACE-FTS measurements are available in January, and consequently the tracer-tracer correlation, artificial tracer and average vortex profile descent techniques could not be applied. As discussed in the introduction, during the winters of 2005/2006, 2008/2009, and 2012/2013 major SSW events and in 2011/2012 strong vortex disturbance occurred (e.g., Manney et al., 2008b, 2009b; Coy et al, 2009; Manney et al., 2015); consequently there were not sufficient measurements inside the polar vortex in March to perform the analysis with ACE-FTS. The ozone depletion inside the Arctic polar vortex was estimated for the remaining winters of 2004/2005, 2006/2007, 2007/2008, 2009/2010, and 2010/2011. Note that the ozone loss estimation for the 2009/2010 winter is the most challenging due to the dynamics and associated strong mixing processes in that year.

## 3 Different estimation methods used for the polar ozone loss

### 3.1 Tracer-tracer method

The tracer-tracer correlation method is based on the assumption that the relationships between long-lived tracers are constant inside an isolated polar vortex (e.g., Proffitt et al., 1993; Müller et al., 2001, 2003; Sankey and Shepherd, 2003; Tilmes et al., 2003, 2004). An empirical relation between a tracer and ozone can be estimated inside the vortex prior to a time when chlorine activation would occur. To derive this correlation function, the polar vortex has to be well established and isolated to limit the influence of mixing processes that could be occurring. In the Arctic, this typically occurs in December or January. This "early vortex reference function" provides the relation between the tracer and ozone in a chemically undisturbed environment. The





passive ozone (that includes dynamical processes only) can then be estimated from the early vortex reference function and the tracer concentration later in spring. The chemical ozone loss is defined as the difference between the observed ozone and the calculated passive ozone based on the simultaneous tracer measurements. The uncertainty of the estimated ozone depletion due to chlorine activation is calculated from the $\pm 1\sigma$ standard deviation of the fitted reference function.

As described in Sect. 2.2, measurements taken in January inside the polar vortex are used to quantify the ozone distribution before significant ozone depletion occurs. This dataset is then compared to measurements taken in March, when chemical ozone depletion is most pronounced in the observed ozone profile. This method has been criticized for neglecting processes that mix extra-vortex air into the polar vortex (e.g., Rex et al., 2002), because it assumes that the polar vortex is isolated, which is not true for all years, especially in the Arctic. By using the sPV criteria described above, we attempt to limit the influence
of mixing of extra-vortex air in our calculation of the early vortex reference function. The tracer-tracer correlation method also neglects descent from high altitudes that invalidates the use of tracer-tracer relationships that include only lower to middle stratospheric data (e.g., Michelsen et al., 1998a, b; Plumb et al., 2000, 2003; Plumb, 2007). Consequently, this could result in a different profile of ozone loss for each tracer.

With the tracer-tracer correlation method, a variety of tracer gases can be used. A tracer is required to be long-lived and
stable (Plumb and Ko, 1992) and thus, not influenced by chemical processes. Since ACE-FTS retrieves profiles for a large number of different trace gases, we have tested six different tracers: $CH_4$, $HF$, $N_2O$, $CCl_3F$, $CCl_2F_2$, and $OCS$. The ACE-FTS measurements in January (black dots) and March (blue dots) for 2011 together with the estimated early vortex reference function (red solid line) are displayed in Fig. 1. There is evidence of large chemical ozone depletion in March, since the March measurements are far from the estimated reference function. The ozone loss is estimated as the difference between the
measurements (blue dots) and the early vortex reference function (red line). For the estimation of the early vortex reference function, a 4th order polynomial was fitted for all of the different tracers. Previous studies have used a 3rd (e.g., Müller et al., 1997) or 4th order polynomial fit (e.g., Tilmes et al., 2003; Müller et al., 2003; Tilmes et al., 2004). We found that at least a 3rd order polynomial is required, with little difference between 3rd and 4th order; 4th order is chosen to be consistent with the more recent publications.

## 3.2  Artificial tracer method

The amount of mixing of extra-vortex air into the polar vortex varies widely depending on the dynamics of each winter and spring, and is more likely to occur in the NH (WMO, 2014). Neglecting mixing processes from the edge of the polar vortex could result in an overestimation of the chemical ozone loss when using the tracer-tracer method, and mixing within the vortex from high altitudes can lead to an underestimation of the chemical ozone loss (e.g., Rex et al., 2002; Müller et al., 2005). One
method that provides a mixing correction, in addition to descent, is the artificial tracer method. This method was first proposed by Esler and Waugh (2002) and uses a "tracer" created from a linear combination of several different trace gases that is linearly correlated with ozone. This linear correlation makes it easier to determine the ozone loss and reduces the impact of mixing, since mixing would only result in "moving" the air parcels along this linear correlation line (Esler and Waugh, 2002). Initially





such an artificial tracer method was used by Esler and Waugh (2002) to estimate denitrification inside the Arctic polar vortex. However, this same method can be applied to estimating the chemical ozone loss as was done by Jin et al. (2006).

Different combinations of tracers can be used to estimate an artificial tracer. Here, we have tested four different combinations that were employed by Esler and Waugh (2002) and Jin et al. (2006). These tracers include a combination of:

1. $N_2O$, $CH_4$, $CCl_3F$, and $CCl_2F_2$ (Esler and Waugh, 2002),

      2. $N_2O$, $CH_4$, $CCl_3F$, and CFC-113 (Esler and Waugh, 2002),

      3. $N_2O$, $CH_4$, and $CCl_2F_2$ (Esler and Waugh, 2002), and

      4. $N_2O$, $CH_4$, OCS, and $CCl_3F$ (Jin et al., 2006).

These artificial tracers will be referred to, in this paper, as Tracer 1, Tracer 2, Tracer 3, and Tracer 4, respectively. To estimate the
artificial tracer that is linearly correlated with ozone, ACE-FTS measurements inside the polar vortex in January are employed. The correlation is then used to estimate the passive ozone levels that would be observed without chemical ozone depletion in March. The difference between the observed ozone and estimated passive ozone equals the chemically depleted ozone between January and March. The linear combination needed to obtain the artificial tracer is estimated for each year, since the trace gas concentrations and the tracer-ozone correlation of these can vary from year-to-year. It is assumed that the linear combination
is constant on a shorter time frame, e.g., within the polar vortex of one winter (Esler and Waugh, 2002). This combination was found to be similar (typically with constants on the same order of magnitude, see supplementary Tables S1 and S2) in some years between 2004 and 2013, but was not the same for each year. The uncertainty of the calculated passive ozone is estimated from the $\pm 1\sigma$ standard deviation of the fitted linear correlation. The total error of ozone loss is derived from the uncertainty of the passive ozone and the ACE-FTS v3.5 statistical fitting error for ozone, which are added in quadrature.

An example of the artificial tracer correlation for all four artificial tracers is shown in Fig. 2. The data shown are mixing ratios inside the polar vortex during January (black dots) and March (blue dots) 2011. While Tracer 1, Tracer 2, and Tracer 4 show a linear correlation with ozone and a small standard deviation, Tracer 3 is not quite linearly correlated and consequently has a larger uncertainty. There is strong evidence in these figures of the chemical ozone depletion in 2011, since the ozone levels in March are very low compared to January and are not linearly correlated with the artificial tracers. The passive ozone
is estimated from the linear fit, by assuming that without any chemical ozone depletion, the ozone levels should still follow this correlation in March. The linear combinations used to estimate the artificial tracers for the 2011 dataset, are:

$$[\text{Tracer1}]_{ppb} = 9.34 \times 10^{-4}[\text{CH}_4]_{ppb} - 7.45 \times 10^{-4}[\text{N}_2\text{O}]_{ppb}$$
$$- 3.41 \times 10^{-3}[\text{CCl}_2\text{F}_2]_{ppt} - 9.46 \times 10^{-3}[\text{CCl}_3\text{F}]_{ppt} + 2.86, \tag{1}$$

$$[\text{Tracer2}]_{ppb} = 4.86 \times 10^{-3}[\text{CH}_4]_{ppb} - 1.63 \times 10^{-2}[\text{N}_2\text{O}]_{ppb}$$
$$+ 1.43 \times 10^{-3}[\text{CFC-113}]_{ppt} - 1.52 \times 10^{-2}[\text{CCl}_3\text{F}]_{ppt} - 0.175, \tag{2}$$



$$[\text{Tracer3}]_{ppb} = 3.20 \times 10^{-4}[\text{CH}_4]_{ppb} - 1.73 \times 10^{-3}[\text{N}_2\text{O}]_{ppb}$$
$$- 5.59 \times 10^{-3}[\text{CCl}_2\text{F}_2]_{ppt} + 3.77, \tag{3}$$

$$[\text{Tracer4}]_{ppb} = 2.22 \times 10^{-4}[\text{CH}_4]_{ppb} - 5.03 \times 10^{-4}[\text{N}_2\text{O}]_{ppb}$$
$$- 3.91 \times 10^{-3}[\text{OCS}]_{ppt} - 6.05 \times 10^{-3}[\text{CCl}_2\text{F}_2]_{ppt} + 3.15. \tag{4}$$

Since Tracer 3 is not highly linearly correlated with ozone ($R^2 = 0.8$, the other tracers have $R^2 \geq 0.9$, see Tables S1 and S2) and has a standard deviation of approximately 10 %, see Fig. 2, this tracer has been eliminated from further analysis as it seems

unsuitable to determine the passive ozone accurately. Tracer 2 contains CFC-113, which has limited coverage at higher altitudes due to a processing issue. As such, limited measurements are available to determine the passive ozone with this artificial tracer. Consequently, Tracer 2 is not a suitable tracer to use with the ACE-FTS v3.5 dataset. For further analysis only Tracer 1 and Tracer 4 were considered for determining the ozone depletion.

### 3.3 Average vortex profile descent technique

Chemical ozone loss can be estimated by applying average vortex profile descent rates to the observed winter ozone profiles. This determines the approximate vortex average passive ozone profile that would be observed without chemical ozone depletion in spring for an isolated vortex. This method has previously been used by, e.g., Manney et al. (2006) and Jin et al. (2006) to estimate Arctic ozone loss. The descent rates between January and March are derived at multiple potential temperature levels from the profile of a long-lived tracer. These descent rates are then applied to the winter ozone profile to determine the passive

ozone profile in March. This method was originally utilized by estimating the average vortex profile descent rate from $\text{N}_2\text{O}$ profiles within the polar vortex, but many tracers can be used for this technique. Here, we have determined the chemical ozone depletion by applying the profile descent rates, between January and March, from six long lived tracers: $\text{CH}_4$, HF, $\text{N}_2\text{O}$, $\text{CCl}_3\text{F}$, $\text{CCl}_2\text{F}_2$, and OCS. Note, this method only allows the estimation of one vortex averaged passive ozone profile; all other methods applied in this study estimate a passive ozone mixing ratio for each data point in March. Consequently, this

method does not consider any changes of the passive ozone levels that can occur throughout March. The uncertainty of the passive ozone is estimated based on the $\pm 1\sigma$ standard deviation of the average vortex profile descent. To obtain the total uncertainty, the statistical fitting error of the ACE-FTS tracer measurements and the uncertainty of the passive ozone are added in quadrature.

An example for the average ACE-FTS $\text{N}_2\text{O}$ profiles inside the polar vortex between January (black line) and March (green

line) 2011 is displayed in Fig. 3 (a). The strongest descent rates occur for high potential temperature levels (approximately -25 K/1.5 months), and very slow descent was observed at 400 K (approximately -4 K/1.5 months). Figure 3 (b) displays the observed ozone in January (black dots) and March (green dots) 2011. The passive ozone profile estimated from the $\text{N}_2\text{O}$ average descent rate between January and March is shown as a blue line. The difference between the observed March ozone





concentrations and passive ozone profile is the estimate of the chemical ozone loss, and is shown as red triangles. Similar figures using the remaining tracers (except for $CCl_3F$ as not enough data were available in 2011) can be found in the supplementary material (Figs. S1-S4). The plots are very similar for all of the tracers used in this study.

### 3.4 Passive subtraction

5 In addition to approaches that only use the ACE-FTS dataset, the chemical ozone depletion was estimated by employing passive ozone from CTMs. The passive ozone from two different models, the Lagrangian ATLAS (Alfred Wegener insTitute LAgrangian chemistry/transport System; Wohltmann and Rex, 2009; Wohltmann et al., 2010) and the Eulerian SLIMCAT (Chipperfield et al., 2006) models, has been used to investigate the chemical ozone depletion in March between 2004 and 2013. Within these models, ozone can be treated as a passive tracer that is not influenced by chemical depletion processes, 10 and only dynamics are applied to the modelled ozone concentrations. In both models, ozone chemistry can be included by employing appropriate chemical reactions.

### 3.4.1 Passive subtraction with ATLAS

The ATLAS model was specifically developed to assess stratospheric chemistry, transport and mixing. Passive ozone and ozone that responds to both heterogeneous and homogeneous chemistry can be estimated with this model, however; in this study only 15 the passive ozone is used and compared to the ACE-FTS measurements to obtain the chemical ozone depletion. This model was previously used to estimate stratospheric ozone within the polar vortex (e.g., Adams et al., 2013; Wohltmann et al., 2013), and validation comparisons with measurements and other models have shown good agreement (Wohltmann and Rex, 2009; Wohltmann et al., 2010, 2013). For the model run presented in this study, the passive tracer was initialized each year on 1 January with Aura MLS (Waters et al., 2006) v3.3/3.4 ozone measurements. The ACE-FTS dataset cannot be used for this 20 since its daily latitude coverage is not sufficient for the initialization of the model. However, relative differences between the Aura MLS and ACE-FTS ozone concentrations are small, between 2 and 5 % in the stratosphere (Sheese et al., 2016). The model was driven by the European Centre for Medium-range Weather Forecasts Reanalysis Interim (ECMWF ERA Interim) meteorological reanalysis (Dee et al., 2011). The passive ozone output has a horizontal resolution of 150 km. The vertical coordinate is potential temperature (∼350 to 1900 K). The vertical resolution of the model changes depending on altitude and 25 is typically between 10 and 40 K at altitudes between 350 and 550 K. Passive ozone concentrations are saved every 12 h at 00:00 UTC and 12:00 UTC.

Since ATLAS is a Lagrangian transport model, the locations of the model output change and are most likely not coincident with the location of the ACE-FTS measurements. To obtain the passive ozone concentration at the location of each 1-km tangent altitude for each ACE-FTS measurement, back or forward trajectories are utilized at individual altitudes to obtain the ACE-FTS 30 measurement location or "end point" at the time of the ATLAS output. Since passive ozone amounts are obtained from ATLAS every 12 h, the back or forward trajectories are estimated for a maximum of 6 h. These forward and back trajectories were calculated with HYSPLIT (HYbrid Single-Particle Lagrangian Integrated Trajectory; Draxler and Hess, 2004), using the NCEP (National Centers for Environmental Prediction) reanalysis (Kalnay et al., 1996) for the meteorological input. Note, the time





period of the back and forward trajectories is relatively short (a maximum of 6 h), therefore differences of the meteorological input used to drive the CTM and the one used for the trajectory calculations are small compared to the total uncertainty. The ATLAS data points that are at the same potential temperature levels (within ATLAS vertical resolution) as the end point of the ACE-FTS measurement are then triangulated. If the three closest ATLAS points that surround the end point of the trajectory

are inside the polar vortex, they are interpolated to this position using a barycentric method.

The passive ozone mixing ratios are compared to the ACE-FTS measurements in January and March. The difference between the March measurements and the passive ozone is considered the chemical ozone loss. The difference between the ACE-FTS dataset and ATLAS for this month is used to estimate an uncertainty of the modelled ozone. To determine the uncertainty of the model results, the relative differences between ACE-FTS measurements and the ATLAS passive ozone for January are calcu-

lated as [ACE-FTS-ATLAS]/[0.5×(ACE-FTS+ATLAS)]. These vary between 0.7 and 5.2 %, depending on the individual year. Note that these uncertainty estimates may include the effects of January ozone loss, which cannot be determined from these datasets. For the total uncertainty of the chemical ozone loss, the statistical fitting error from ACE-FTS v3.5 $O_3$ measurements and the mean difference of ACE-FTS measurements and ATLAS passive ozone in January are added in quadrature.

An example of the comparisons for January 2011 is shown in Fig. 4 (a). ATLAS passive and ACE-FTS measured ozone are

in good agreement. The difference is on average $-5.2 \pm 0.7$ % with a high correlation coefficient ($R = 0.94$). This difference between the ATLAS passive and ACE-FTS measured ozone is likely due to the difference between the Aura MLS and ACE-FTS datasets that is of the same order of magnitude. However, some of this difference could also be due to early ozone depletion in January, as was seen by Manney et al. (2015). The ACE-FTS measurements (green dots) and ATLAS passive ozone (blue dots) for March 2011 are displayed in Fig. 4 (b). The difference between the ATLAS and ACE-FTS ozone concentrations are

displayed as red triangles and indicate chemical ozone loss.

### 3.4.2  Passive subtraction with SLIMCAT

In addition to ATLAS, we have also used the SLIMCAT off-line 3-D CTM to investigate Arctic ozone loss. This model has been widely used to study the stratospheric ozone abundance and chemical ozone depletion (e.g., Feng et al., 2007; Sinnhuber et al., 2000; Singleton et al., 2005, 2007; Adams et al., 2012; Lindenmaier et al., 2012; Dhomse et al., 2013; Chipperfield et

al., 2015). A detailed description of the model can be found in Chipperfield et al. (2006) and recent updates are described in Dhomse et al. (2013) and Chipperfield et al. (2015). SLIMCAT uses an Eulerian grid that extends from pole to pole. It contains a detailed stratospheric chemistry scheme including all processes that are related to polar ozone depletion. As such, passive ozone and ozone that responds to ozone chemistry are modelled. The model was also forced by ERA-Interim meteorological reanalysis (Dee et al., 2011). The passive ozone from the SLIMCAT model run was reset on 1 January for each year. The

model simulation used here has a horizontal resolution of $2.8° \times 2.8°$ and the vertical coordinate is defined on hybrid $\sigma$-pressure vertical levels between the surface and approximately 60 km on 32 layers. The simulation was initialised in 1979 (using output from a 2-D model) and constrained by specified global mean surface observations of long-lived source gases. SLIMCAT therefore simulates ozone and all other stratospheric trace gases for all years in this study in a single long-term simulation. The model was sampled at the locations of the 30 km tangent altitude of the ACE-FTS occultations providing





profiles of the passive ozone and ozone that responds to ozone chemistry corresponding to each measurement. Although the geo-location of the ACE-FTS measurements change with altitude, the location of the measurements at the altitudes of interest (approximately 15-25 km) are up to approximately $0.5°$ of the location of the 30 km tangent altitude and, therefore, within the model resolution. Therefore, the measurements and the modelled ozone fields can be directly compared without further

processing. The ozone loss was estimated from the difference between the modelled passive ozone and the observed ozone inside the polar vortex in March. Additionally, the ozone loss has also been estimated by solely using both modelled ozone that responds to ozone chemistry and passive ozone from the model (referred to as "SLIMCAT only"). This helps to estimate the uncertainty of the modelled ozone (that includes ozone chemistry) by comparing it to the measurements, and can indicate potential ozone loss in January by comparing the passive and ozone (that includes ozone chemistry). To estimate the uncertainty

of the model results, the relative differences between ACE-FTS measurements and the SLIMCAT ozone for January and March are calculated as [ACE-FTS-SLIMCAT]/[0.5×(ACE-FTS+SLIMCAT)]. The ACE-FTS ozone measurements and the modelled ozone agree very well: with mean relative differences between 0.8 and 4.8 % (and $R \approx 0.95$), depending on the specific year. The total uncertainty of the ozone loss was estimated in a similar way as was done for the ATLAS analysis (see Sect. 3.4.1), the ACE-FTS ozone measurement fitting error and the mean relative difference between ACE-FTS and SLIMCAT ozone were

added in quadrature.

An example of the comparison for January and March 2011 is shown in Fig. 5 (a). The measurements and the model ozone are in good agreement with a mean difference of $3.9 \pm 0.8 \%$ and the correlation is high with a correlation coefficient $R = 0.95$. This result confirms that the model simulates the measured ozone quite well. In Fig. 5 (b) and (c), ACE-FTS measurements (green dots), SLIMCAT ozone (cyan triangles) and SLIMCAT passive ozone (blue dots) are displayed for January and March

2011, respectively. The ozone loss (red triangles) is obtained from the observed and modelled passive ozone, and indicate the chemical ozone loss. Figure 5 (b) confirms that little ozone depletion was observed in January 2011, as the differences between the measured and modelled passive ozone are on average around 0.1 ppmv. The results of the estimated ozone loss are discussed in Sect. 4.

## 4 Annual intercomparison and interpretation of Arctic ozone loss estimates

In this section, the impact of the different tracers and the different methods on the estimated ozone loss is discussed for the five years where no SSW event occurred. The mixing ratio profile and partial column (380 K - 550 K) ozone depletion are compared and the differences are discussed.

### 4.1 Impact of the choice of tracer

For the tracer-tracer correlation method and the average vortex profile descent technique, six long-lived tracers ($CH_4$, $N_2O$,

HF, OCS, $CCl_3F$, and $CCl_2F_2$) have been used to estimate the chemical ozone depletion in the Arctic polar vortex in March with respect to January. These results are shown in Figs. 6 and 7, respectively. Two different combinations of tracers have been investigated to create an artificial tracer that is linearly correlated with ozone, and these results are displayed in Fig. 8. Panel (a)



of Figs. 6-8 shows the mixing ratio loss profile of the maximum chemical ozone loss between 380 K and 550 K in March 2005, 2007, 2008, 2010, and 2011. The partial column ozone loss presented here is estimated from the mixing ratio losses using the mean altitudes of the DMP potential temperature profile between 380 K and 550 K, and the ACE-FTS densities at the given altitude level. This interpolation to altitude levels was necessary for the estimation of the integrated partial columns (Livesey et al., 2015, personal communication in 2016, N. Livesey). Panels (b) and (c) of Figs. 6-8 show the maximum and mean partial column ozone loss, respectively. The error bars displayed in panels (b) and (c) of Figs. 6-8 indicate the uncertainty of the maximum and mean column ozone loss estimate that are derived from maximum and mean ozone loss VMR uncertainties, respectively, as calculated in Sect. 3.

For the tracer-tracer correlation method (Fig. 6), the results for all six tracers are similar for the partial column ozone. However, there are differences apparent in the profile of the estimated ozone loss for each tracer, especially for high and low altitudes. The estimated uncertainties are ~0.2-0.6 ppmv, or ~10-20 % and the results from all tracers agree within the uncertainties between approximately 460-500 K for all years, with the exception of 2005. Both, mixing and strong ozone loss was apparent in the winter of 2004/2005 (e.g., Manney et al., 2006), and is consequently a good year to test the agreement between the different tracers. As shown in Fig. 6 (a), the profiles of the different tracers do not agree well in March 2005. This indicates the failure of the tracer-tracer correlation method, even though only inner core vortex measurements were used for estimating the ozone loss. These results are in agreement with the discussions of the tracer-tracer correlation method in previous studies (e.g., Michelsen et al., 1998a, b; Plumb et al., 2000, 2003; Plumb, 2007) that further confirmed the tracer-tracer correlation method to be inaccurate for estimating Arctic ozone loss.

Furthermore, both OCS and $CCl_3F$ results show a smaller ozone loss ($\sim$ 0.5-1 ppmv) above approximately 500 K compared to the other tracers in all years. For most years, the ozone loss profiles computed with $CH_4$, $N_2O$, HF, and $CCl_2F_2$ agree well and within the estimated uncertainties for the entire profile. The largest discrepancies between the tracers occur in 2005, when the vortex was relatively weak and influenced by mixing. Also, in 2007, the estimated loss is larger if HF is used as a tracer, likely because of mixing. For the partial column losses, all tracers agree within the estimated uncertainties. However, these uncertainties are quite large, between approximately 20 DU and 40 DU, which represent roughly 40-60 % of the estimated ozone loss. The estimated profile and partial column ozone loss is consistently smaller ($\sim$ 10 DU) if OCS or $CCl_3F$ is used as the tracer. In the ACE-FTS v3.5 dataset many $CCl_3F$ retrievals fail, especially in higher altitudes. Typically only one quarter to half as many profiles are available each year compared to the other tracers. Due to this limited coverage, the column ozone loss could only be estimated with $CCl_3F$ in 2011. It should be noted that OCS has a significantly shorter stratospheric lifetime that is approximately 2 years (Montzka et al., 2007; Dhomse et al., 2014), whereas all other tracers have lifetimes of 50+ years (Hoffmann et al., 2014; Brown et al., 2013). As OCS is not as stable as all other tracers, this could negatively impact the ozone loss estimation using OCS.

Using these six different tracers to estimate the average vortex descent rate (Fig. 7) leads to very similar results for most tracers, except for OCS. The uncertainties for this method are ~0.02-0.1 ppmv, or ~1-10 %. These are smaller than the ones estimated for the tracer-tracer method due to the small standard deviation of the average vortex descent profile. During the winters of 2004/2005, 2006/2007, and 2009/2010 when using OCS, ascent inside the polar vortex rather than descent is



estimated (for all calculated descent rates, see Supplementary Tables S3-S8). In 2007/2008 and 2010/2011, when $CH_4$, $N_2O$, HF, and $CCl_2F_2$ estimate large descent of approximately 20-35 K over 1.5 months (between approximately 450-550 K, see Supplementary Tables S3-S8), OCS only estimates half as much. The reason for this could be the limited precision of the ACE-FTS OCS retrievals that have retrieval fitting errors of around 10 %, almost 10 times higher than for other species (e.g., $O_3$ and

$N_2O$). As shown in Fig. 7 (a), the mixing ratio loss profile is very similar for all different tracers, with the exception of HF in 2007 where a larger chemical ozone depletion is estimated. This large discrepancy is also seen in 2007 for the tracer-tracer correlation method. For this winter, the descent rates using HF are almost twice as large as those derived from the other tracers; for all other years HF provides descent rates that are similar to the other tracers. Due to the large estimated uncertainties of the integrated loss ($\sim 2.4 - 6.5$ DU), the estimated partial column ozone loss for each year agrees for all different tracers within

the estimated uncertainties. Only in 2010 could the partial column ozone depletion (between 380-550 K) be estimated using $CCl_3F$ because of limited retrievals for this species. Because ACE-FTS fitting errors of OCS are quite large and there are limited ACE-FTS $CCl_3F$ retrievals, it is not advised to use OCS or $CCl_3F$ with the ACE-FTS v3.5 dataset to determine the average vortex profile descent rates and subsequent ozone loss.

Two different combinations of tracers have been used to create artificial tracers that are used to estimate ozone losses

(Fig. 8). The ozone loss profiles and partial column ozone losses computed from Tracer 1 and Tracer 4 are very similar, with differences of typically less than 0.1 ppmv (the uncertainty of these calculated ozone loss profiles is 0.2-0.3 ppmv at all altitudes). As expected based on the profile ozone loss, the estimated column ozone losses shown in Figs. 8 (b) and (c) agree within approximately 5 DU for Tracer 1 and Tracer 4. The uncertainties of the column ozone loss for these artificial tracers are typically between 15 DU and 30 DU (this equals roughly 30-50 % of the estimated loss depending on the year).

To summarize, for the tracer-tracer correlation, average vortex profile descent, and artificial tracer correlation approaches, the computed partial column ozone losses for most tracers agree within the estimated uncertainties (which can vary between approximately 2.5 and 40 DU depending on the method and year). The results for the profile and partial column losses are most consistent when using $CH_4$, $N_2O$, HF, and $CCl_2F_2$ for the tracer-tracer correlation and profile descent techniques. However, using OCS or $CCl_3F$ as a tracer seems to result in larger uncertainties and has the disadvantage that there are not as many

profiles available as there are for the rest of the tracers. Based on this analysis with the ACE-FTS v3.5 dataset, the best choices of tracers are $CH_4$, $N_2O$, HF, and $CCl_2F_2$. For the artificial tracer method, Tracer 1 and Tracer 4 seem to be equally suited for ACE-FTS v3.5.

To be able to compare between the different methods using the tracer-tracer correlation, the artificial correlation and the average profile descent techniques in the following section, average mixing ratios and integrated ozone losses have been

calculated as follows. The average mixing ratio loss using $CH_4$, $N_2O$, HF, and $CCl_2F_2$ are utilized for the tracer-tracer correlation and average vortex profile descent methods. For the artificial tracer method, the average of Tracer 1 and Tracer 4 is employed. The uncertainties of these averages have been computed by propagating the error from each method and tracer. Note, the higher losses using HF as a tracer in 2007 increase the partial column and the profile ozone loss by approximately 7 % ($\sim$3 DU and $\sim$0.05 ppmv, respectively).



## 4.2 Comparison between the different methods

The mixing ratio and partial column ozone losses have been derived for March 2005, 2007, 2008, 2010, and 2011 using six different methods, as described above, and are shown in Fig. 9. As expected, all of these methods consistently show the greatest chemical ozone loss in March 2011 (e.g., Manney et al., 2011; WMO, 2014; Livesey et al., 2015). The second largest ozone

depletion event for these years occurred in 2005. Based on our results, the losses in 2007 and 2008 seem to be similar and are only slightly smaller than those in 2005. In 2010, the mean partial column ozone depletion seems to be lower than for the other years. This can be explained by mixing and the break up of the polar vortex during this winter (e.g., Dörnbrack et al., 2012; Kuttippurath and Nikulin, 2012; Wohltmann et al., 2013). These estimated losses are as expected and are consistent with previous studies (WMO, 2011, 2014; Livesey et al., 2015).

The maximum mixing ratio loss profiles, as displayed in Fig. 9 (a), show reasonably good agreement between the different methods. The maximum ozone losses computed at 460 K (the approximate height of the peak ozone loss) are 1.2-1.6 ppmv (mean: 1.5 ppmv) in 2005 (excluding the tracer-tracer method that showed a peak loss of 1.9-2.5 ppmv), 1.2-1.8 ppmv (mean: 1.5 ppmv) in 2007, 1.1-1.4 ppmv (mean: 1.3 ppmv) in 2008, 0.9-1.3 ppmv (mean: 1.1 ppmv) in 2010, and 2.1-2.7 ppmv (mean: 2.3 ppmv) in 2011. The uncertainties of these losses are on the order of 5-10 % for all methods, except for the average vortex

profile descent technique that is around $\pm 1.5\,\%$. There is significantly larger estimated ozone loss in 2005 using the tracer-tracer correlation method. The estimated peak ozone loss at 460 K is $2.2 \pm 0.3$ ppmv whereas the other methods estimate the loss between $1.2 \pm 0.1$ ppmv and $1.6 \pm 0.1$ ppmv. As previously discussed, the Arctic stratosphere in 2005 was affected by strong ozone loss and mixing (WMO, 2006), and is consequently an ideal year to test whether the different methods agree and the models are accurate (Livesey et al., 2015). The tracer-tracer correlation method and the average vortex descent technique differ

significantly from all other estimation methods. These discrepancies highlight the difficulties using the tracer-tracer correlation method because mixing processes and descent in the 2005 Arctic vortex are not considered, and the difficulties using the average vortex descent technique in years of an unstable polar vortex. This can therefore lead to an overestimated ozone loss using the tracer-tracer correlation method and an underestimated ozone loss using the average vortex descent technique. In all other years, the tracer-tracer correlation method agrees well with the other five methods. The average vortex descent technique

agrees well with the other methods in 2007, and 2008.

The mean and maximum partial column ozone losses are summarized in Table 1 and are displayed in Figs. 9 (b) and (c), respectively. The uncertainties from the tracer-tracer correlation and artificial tracer methods are large in all years compared to all other methods. These are on the order of 10-20 DU and are based on the $\pm 1\sigma$ standard deviation of the early vortex reference function (see Sect. 3.1 and 3.2 ). Much smaller uncertainties (2-10 DU) have been determined for calculations using the model

output, which are based on the mean differences between the measurements and model output. For ATLAS, this is based on the difference between the model passive ozone and measurements in January each year. In contrast, the ozone loss uncertainty computed with SLIMCAT is based on the difference between the model ozone and measurements in both March and January, and is very similar to the estimated ATLAS uncertainties. The maximum ozone loss in March combining all mehods (see Fig. 9 (b) and Table 1) was estimated to be 86 DU in 2005, 76 DU in 2007, 72 DU in 2008, 59 DU in 2010, and 109 DU in 2011.



The March mean ozone loss (Fig. 9 (c) and Table 1) obtained from these methods is 57 DU, 44 DU, 52 DU, 30 DU, and 66 DU for 2005, 2007, 2008, 2010, and 2011, respectively.

Discrepancies are apparent between the measurement only and the passive subtraction methods in 2010, especially for the computed mean partial column loss. Each time the vortex splits and the two parts reunite, extra-vortex air is mixed, therefore, for 2010, the tracer-tracer and the profile descent techniques are not reliable since an isolated vortex is essential for these methods that do not account for the mixing of extra-vortex air. The results of the artificial tracer technique should be uninfluenced by mixing. The loss estimates in 2010 using the artificial tracer technique do not agree with the passive subtraction methods. It is worth noting that the passive subtraction methods compute similar losses from year to year, including 2010, when the vortex was much disturbed. The passive subtraction methods may smooth out the year-to-year differences and model results in some years may compute some ozone loss even in the absence of $ClO_x$ chemistry.

The maximum ozone loss computed from the average vortex profile descent technique is low compared to the other methods; however, the mean losses agree. This discrepancy is likely because the the average profile descent technique only provides a mean passive ozone profile. Hence, this method is capable of estimating an average across the vortex but not a specific maximum loss. Using modelled passive ozone to determine the mean loss leads to larger ozone loss than for the methods that are using measurements only. This may be in part because the models are initialized on 1 January each year, whereas ACE-FTS measurements start at the end of January. However, based on the difference between the modelled passive ozone (from ATLAS and SLIMCAT) and measured ozone in January, this can only account for a difference of up to 6 %.

Using the passive subtraction method with a Lagrangian (ATLAS) or an Eulerian (SLIMCAT) model leads to very similar computed ozone losses. For the maximum partial column ozone loss, the results from both models agree to within the estimated uncertainties for all years. These differences are between 1 and 9 DU (between 2 and 12 %), where the smallest difference occurs in 2011, and the largest in 2008. The mean partial column ozone losses agree within 1-15 DU, and are within the estimated uncertainties for most years (except in 2005). The ozone loss has also been estimated using only the SLIMCAT ozone and passive ozone ("SLIMCAT only"). The mean loss results for using SLIMCAT passive ozone and ACE-FTS measurements are very similar to the mean losses computed from SLIMCAT only and differ between 0.3-8.0 DU (0.5-14 %); the largest difference was found in 2007. The maximum losses are also similar between these two estimation methods. These differences are between 2 and 8 DU, and the largest difference (8 DU, $\sim$ 13 %) was found in 2010, a year when the polar vortex was highly influenced by mixing, that is still within the estimated uncertainties.

Other studies have used the Aura MLS dataset to look at ozone depletion over this time period. Between 1 January and 1 April, for the years studied, Livesey et al. (2015) found losses of 1 ppmv at 450 K in 2005, 2007, and 2008, and around 2 ppmv in 2011 using a Match-based approach that uses trajectory calculations to identify the same air parcel measured at various times (von der Gathen et al., 1995; Rex et al., 1998). During the same time period, Kuttippurath et al. (2010) and Kuttippurath et al. (2012), who used a passive subtraction approach, found peak losses at approximately 450-475 K of 1.5 ppmv in 2005, 1.2 ppmv in 2007, 1.4 ppmv in 2008, 0.9 ppmv in 2010, and 2.4 ppmv in 2011, respectively. The mixing ratio losses estimated (excluding the previously discussed outlier, the tracer-tracer correlation method in 2005) in this study agree well with Livesey et al. (2015) and Kuttippurath et al. (2010, 2012). Here, we found similar losses at around 460 K, 1.2-2 ppmv in 2005, 1.0-1.5 ppmv in 2007,



1.2-1.6 ppmv in 2008, 0.8-1.3 ppmv in 2010, and 2.0-2.7 ppmv in 2011. A comparison of the partial column ozone loss with these two studies is shown in Table 1. It should be noted that the time period used by Livesey et al. (2015) and Kuttippurath et al. (2010, 2012) is slightly longer (1 January to 1 April), including the loss throughout January, and the altitude range is slightly larger; hence, the loss is expected to be larger than the here estimated mean ozone loss columns. Although Livesey

et al. (2015) and Kuttippurath et al. (2010, 2012) have reported the total ozone loss by the beginning of April, our estimated mixing ratio and the partial column ozone losses are consistent with these two studies for all these years. This suggests that not only are the computed losses consistent when using the ACE-FTS dataset with different methods, but also similar ozone losses are computed when the MLS dataset is employed instead. The following section will discuss the ozone depletion in 2011 further as a case study.

Overall, we have found that the different methods agree in most years within the estimated uncertainties considering the profile mixing ratio loss, as well as the mean and maximum partial column ozone loss. Typically, the average vortex profile descent method estimates smaller ozone losses compared to all other methods. The tracer-tracer correlation and the artificial tracer approaches have estimated uncertainties that are approximately twice as large compared to the passive subtraction and the average vortex profile descent techniques. This is due to the large uncertainties for the early vortex reference function used

for the tracer methods. Furthermore, consistent results were found using the passive subtraction method with both a Lagrangian and an Eulerian model. For the presented years, the ACE-FTS measurements and the SLIMCAT ozone (that includes ozone chemistry) results are in very good agreement with mean differences of less than 5 % in January and March. As such, similar ozone losses are computed when only SLIMCAT and no measurements are utilized.

### 4.3  Comparison to previous estimates of the 2011 Arctic ozone loss

Since the ozone loss in the Arctic during the 2010/2011 winter was extreme, this particular winter has been widely studied. Therefore, a more comprehensive comparison is available for this specific winter. Table 2 shows the estimated peak chemical ozone depletion and the altitudes at which these losses occurred from these studies. In these previous studies, the passive subtraction (using CTMs) and the Match approaches have been used to determine the Arctic chemical ozone depletion in 2010/2011. Several different instruments, such as Aura MLS, MIPAS and ozonesondes have been employed in these estimates.

The various methods that have been utilized in the current study consistently show the peak of the ozone loss at 460 K. This is in very good agreement with the all of the previous results in Table 2 where the altitude of the peak ozone loss was determined between 450 K and 475 K. The maximum loss simulated in this study at 460 K is between 2.1 ppmv and 2.7 ppmv which agrees well with Sinnhuber et al. (2011), Manney et al. (2011), Kuttippurath et al. (2012), von Hobe et al. (2013), and Livesey et al. (2015), see Table 2. The smallest ozone loss in any of the studies, approximately 2.0 ppmv, was found by Livesey

et al. (2015); when their calculations were updated using MERRA-2 for the trajectory calculations, ozone loss estimates were more in line with those in the other studies (N. Livesey, personal communication in 2016). Of the six methods we used, it was found that the smallest losses are computed for the average vortex profile descent technique from this study. The passive subtraction method has also been used in 2011 by, e.g., Sinnhuber et al. (2011) and Kuttippurath et al. (2012) with different models and datasets than used in this paper. Those results are in good agreement and differ by less than 0.1 ppmv, well within





the estimated uncertainties, from our results using the passive subtraction methods with ATLAS and SLIMCAT. Agreement with these previous studies indicates that our estimated ozone losses are reasonable. Finally, we also conclude that the Arctic ozone loss estimates in 2010/2011 are very similar regarding the peak loss altitude and the mixing ratio loss (with maximum differences of 0.5 ppmv) when various instruments, models and different approaches are utilized.

## 5 Summary and conclusions

This study evaluated the springtime Arctic ozone depletion estimated from various methods over a nine year period between the winters of 2004/2005 and 2012/2013 using the space-borne ACE-FTS dataset. These estimation methods include tracer-tracer correlation, the artificial tracer, the average vortex profile descent, and passive subtraction with ATLAS and SLIMCAT. The chemical ozone depletion was estimated for the Arctic winters of 2004/2005, 2006/2007, 2007/2008, 2009/2010, and 2010/2011. During all other Arctic winters (2005/2006, 2008/2009, 2011/2012, and 2012/2013), the Arctic lower stratospheric vortex was disturbed enough that insufficient measurements were recorded inside the polar vortex in March to estimate ozone loss from ACE-FTS measurements.

ACE-FTS provides retrievals of over 30 trace gases; from these six long-lived tracers were used with the tracer-tracer correlation and average vortex profile descent techniques. We have shown that $CH_4$, $N_2O$, HF, and $CCl_2F_2$ perform equally well for these methods. Using OCS or $CCl_3F$ as tracers for these approaches has shown the following issues: with ACE-FTS OCS, positive descent rates or descent rates that were only half that of the other tracers have been determined, likely due to the limited accuracy of the ACE-FTS OCS retrievals, which have high retrieval fitting errors. Therefore, OCS is not recommended for use as a tracer with the ACE-FTS v3.5 dataset to derive Arctic polar ozone loss. ACE-FTS $CCl_3F$ has limited coverage compared to other species, and this was not sufficient to estimate ozone depletion for each year. Also, four different artificial tracers that linearly correlate with ozone were investigated. Two artificial tracers were identified as suitable tracers for estimating ozone depletion. We found that the combination of $N_2O$, $CH_4$, $CCl_3F$, and $CCl_2F_2$ (Tracer 1), and $N_2O$, $CH_4$, OCS, and $CCl_3F$ (Tracer 4) work equally well.

Comparisons of the ozone loss estimates from the methods in this study with those obtained from other methods and instruments are in good agreement. This is especially the case for the Arctic winter of 2010/2011, which shows a peak ozone loss of 2.0-2.65 ppmv at 450-475 K throughout various methods and datasets. Also consistent with previous studies, strong losses were computed for the 2004/2005. Similar March average ozone losses were seen in March 2007 and 2008, and smaller losses in March 2010 (as shown in Table 1).

Overall, we showed that with one dataset and several ozone loss estimation methods, losses are determined that agree within the estimated uncertainties. The results of the partial column ozone losses from different methods that have been investigated are, for the most part, within the estimated uncertainties (which are, however, quite large for the correlation methods), except for the maximum loss using the average vortex profile descent technique, which is consistently lower than the five other methods shown in this study. While similar ozone losses were computed for all methods in years with an isolated polar vortex, the passive subtraction methods using either ATLAS or SLIMCAT seem to have smaller computed uncertainties. The tracer-tracer





correlation and artificial tracer techniques have large uncertainties because of the large standard deviation of the early vortex reference function. The estimated partial column loss uncertainties for the former are approximately twice as large as estimated with the passive subtraction and the average vortex profile descent techniques. For a highly disturbed vortex and little to no $ClO_x$ activation (e.g., 2010), the passive subtraction methods indicate larger ozone loss and might smooth out the year-to-year

variability. Very little difference was found between using the passive subtraction method with passive ozone from a Lagrangian (ATLAS) and from an Eulerian model (SLIMCAT). For the first time, an evaluation has been performed of these six different ozone loss estimation methods with one dataset. Using the dataset from the space-borne ACE-FTS, we found consistency and good agreement between all methods for winters with a strong and isolated polar vortex. This analysis shows that from the six different estimation methods presented, either the artificial tracer correlation technique or the passive subtraction method (with

ATLAS or SLIMCAT) is best suited for estimating the ozone loss in the Arctic polar vortex. Based on this study, for years with significant $ClO_x$ activation either the passive subtraction or the artificial tracer technique are best suited. For years with little to no $ClO_x$ activation the artificial tracer correlation technique might be the most reliable because it considers mixing and seems to compute a reasonably small ozone loss.

**Data availability**

The ACE-FTS Level 2 dataset used in this study can be obtained via the ACE-FTS website (registration required), http://www.ace.uwaterloo.ca (ACE-FTS, 2017), or upon request from the corresponding author (kaley.walker@utoronto.ca). SLIMCAT output analysed in this current study is available from the corresponding author (kaley.walker@utoronto.ca) or Martyn Chipperfield (m.chipperfield@leeds.ac.uk) on reasonable request. The ATLAS source code is available on the AWIForge repository (https://swrepo1.awi.de/). Access to the repository is granted on request. ATLAS output analysed in this current

study is also available upon request from the corresponding author (kaley.walker@utoronto.ca).

**Competing interests**

The authors declare that they have no competing interests.

*Acknowledgements.* This work was supported by grants from the Canadian Space Agency (CSA) and the Natural Sciences and Engineering Research Council of Canada (NSERC). The Atmospheric Chemistry Experiment (ACE), also known as SCISAT, is a Canadian-led mission

mainly supported by CSA and NSERC. We thank ECMWF for providing reanalysis data. The research leading to these results has received funding from the European Community's Seventh Framework Programme (FP7/2007-2013) under grant agreement no. 603557 (Stratoclim). The SLIMCAT modelling was supported by Natural Environment Research Council (NERC), National Centre for Earth Observations (NCEO) and National Centre for Atmospheric Research (NCAS). The SLIMCAT simulations were performed on the Archer and University of Leeds (Arc2 and N8) HPC systems. M. P. Chipperfield was supported by a Royal Society Wolfson Merit Award. The authors thank Peter

Bernath for his leadership of the ACE mission and Chris Boone for his development of the ACE retrieval.



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

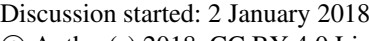



**Table 1.** Maximum and mean partial column ozone loss estimate (in DU) between 380-550 K from various methods (see details in the text), for March 2005, 2007, 2008, 2010, and 2011. The $\pm 1\sigma$ uncertainties are stated and the derivation is described in the text. These results are compared to previous quantifications of the integrated ozone loss (350-550 K) between 1 January and 1 April using the Aura MLS data set by Livesey et al. (2015) and Kuttippurath et al. (2010, 2012).

| Partial column losses | 2005 | 2007 | 2008 | 2010 | 2011 |
|---|---|---|---|---|---|
| Tracer-tracer (max.) | $107 \pm 20$ | $84 \pm 10$ | $80 \pm 15$ | $61 \pm 15$ | $112 \pm 11$ |
| Artificial tracer (max.) | $76 \pm 21$ | $65 \pm 14$ | $62 \pm 13$ | $41 \pm 13$ | $103 \pm 10$ |
| Descent (max.) | $66 \pm 4$ | $61 \pm 5$ | $59 \pm 4$ | $41 \pm 4$ | $85 \pm 2$ |
| ATLAS (max.) | $87 \pm 3$ | $83 \pm 4$ | $82 \pm 8$ | $78 \pm 8$ | $116 \pm 11$ |
| SLIMCAT (max.) | $92 \pm 4$ | $78 \pm 2$ | $73 \pm 5$ | $70 \pm 10$ | $115 \pm 9$ |
| SLIMCAT only (max.) | $86 \pm 3$ | $83 \pm 1$ | $75 \pm 5$ | $62 \pm 7$ | $122 \pm 8$ |
| Average for all (max.) | 86 | 76 | 72 | 59 | 109 |
| Tracer-tracer (mean) | $59 \pm 20$ | $43 \pm 10$ | $52 \pm 15$ | $15 \pm 14$ | $60 \pm 11$ |
| Artificial tracer (mean) | $47 \pm 20$ | $28 \pm 14$ | $44 \pm 13$ | $8 \pm 10$ | $56 \pm 10$ |
| Descent (mean) | $47 \pm 4$ | $39 \pm 5$ | $45 \pm 4$ | $13 \pm 3$ | $61 \pm 2$ |
| ATLAS (mean) | $52 \pm 3$ | $46 \pm 3$ | $56 \pm 7$ | $49 \pm 8$ | $66 \pm 10$ |
| SLIMCAT (mean) | $67 \pm 3$ | $51 \pm 2$ | $56 \pm 5$ | $46 \pm 6$ | $73 \pm 7$ |
| SLIMCAT only (mean) | $67 \pm 3$ | $59 \pm 1$ | $60 \pm 5$ | $51 \pm 7$ | $77 \pm 8$ |
| Average for all (mean) | 57 | 44 | 52 | 30 | 66 |
| Livesey et al. (max.*) | 88 | 54 | 66 | 44 | 117 |
| Kuttippurath et al. (max.*) | 81 | 62 | 90 | 42 | 115 |

*: Integrated loss between 1 January and 1 April, this is approximately equivalent to the maximum losses reported in this study.





**Table 2.** Estimates of the peak altitude and maximum ozone loss observed during the 2010/2011 Arctic winter.

| Study | Method/Instrument | Peak altitude (K) | Loss (ppmv) |
|---|---|---|---|
| This work | Tracer-tracer/ACE-FTS | 460 | $2.28 \pm 0.15$ |
| This work | Artificial tracer/ACE-FTS | 460 | $2.16 \pm 0.14$ |
| This work | Descent/ACE-FTS | 460 | $2.13 \pm 0.03$ |
| This work | Passive subtraction (ATLAS)/ACE-FTS | 460 | $2.46 \pm 0.18$ |
| This work | Passive subtraction (SLIMCAT)/ACE-FTS | 460 | $2.50 \pm 0.12$ |
| This work | Passive subtraction (SLIMCAT only) | 460 | $2.57 \pm 0.12$ |
| This work | Average for all | 460 | 2.34 |
| Livesey et al. (2015) | Match/Aura MLS | 450 | $2.0 \pm 0.3$ |
| Kuttippurath et al. (2012) | Passive subtraction/Aura MLS | 475 | $\sim 2.4$ |
| Manney et al. (2011) | Aura MLS, $O_3$ sonde | 450 | 2.3-2.5 |
| Sinnhuber et al. (2011) | Passive subtraction/MIPAS | 450-475 | 2.3-2.5 |
| von Hobe et al. (2013) | Match/$O_3$ sonde | 460 | $2.6 \pm 0.5$ |

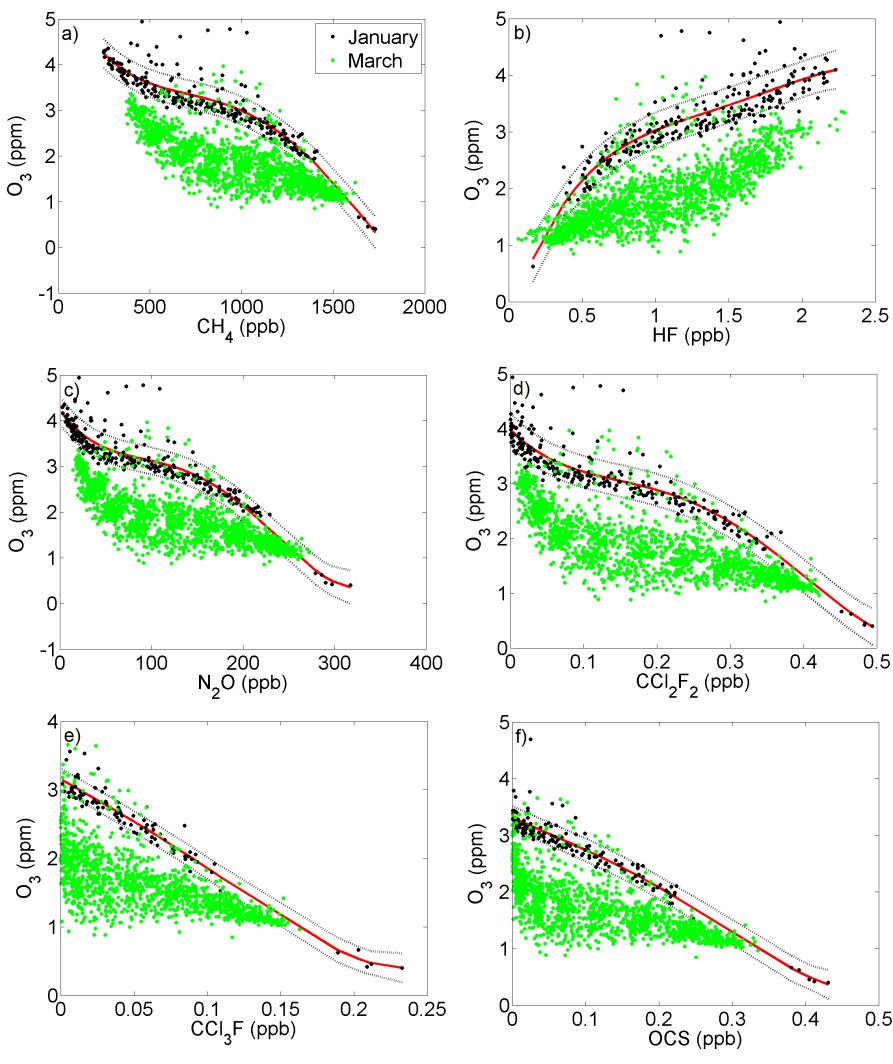

**Figure 1.** $O_3$-tracer correlation using ACE-FTS measurements inside the polar vortex in January (black dots) and March (green dots) 2011 using (a) $CH_4$, (b) HF, (c) $N_2O$, (d) $CCl_2F_2$, (e) $CCl_3F$, and (f) OCS as a tracer, in units of volume mixing ratios. The red line shows the estimated early vortex reference function (see text for details) and the dashed black lines indicate the $\pm 1\sigma$ standard deviation of the fit.



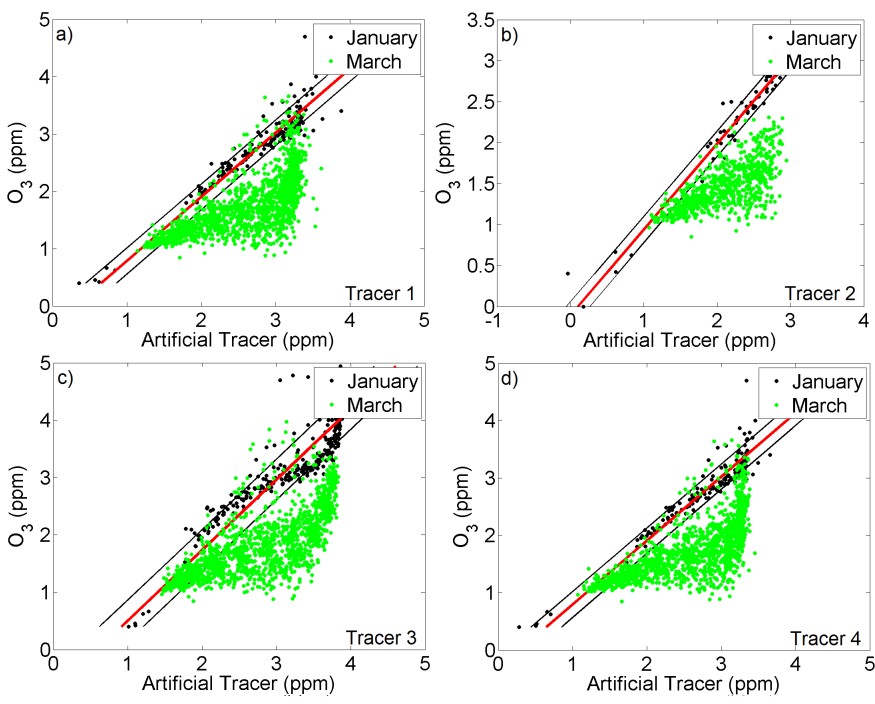

**Figure 2.** Artificial correlation technique using ACE-FTS measurements inside the polar vortex in January (black dots) and March (green dots) 2011 using (a) Tracer 1, (b) Tracer 2, (c) Tracer 3, and (d) Tracer 4. The fitted correlations are shown as red lines, and the black lines indicate the $\pm 1\sigma$ standard deviation of the fit. See text for further details on the artificial tracers.



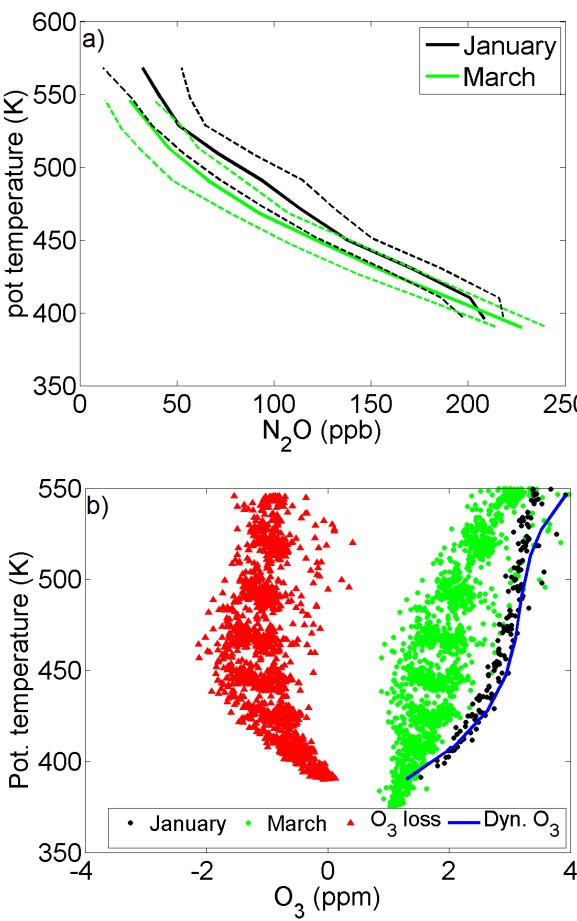

**Figure 3.** Panel (a) shows the monthly average $N_2O$ profiles observed by ACE-FTS inside the polar vortex in January (black line) and March (green line) 2011 together with the respective standard deviations (shown as dashed lines). Panel (b) displays the observed ACE-FTS ozone in January (black dots) and March (green dots) 2011, the passive ozone (blue line) for March 2011, estimated from the average vortex profile descent from $N_2O$, and the ozone loss (red triangles; the difference between observed and average passive ozone in March).





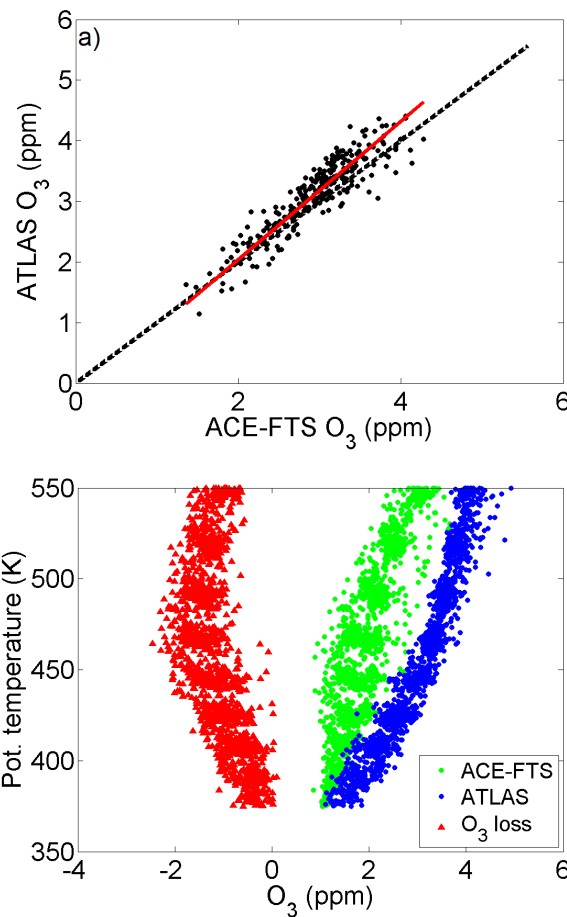

**Figure 4.** Panel (a) shows a comparison between the ATLAS passive ozone and ACE-FTS ozone dataset inside the polar vortex for January 2011. The black dots represent the individual data points and the red line indicates the line of best fit. For easy comparison, the 1-to-1 line is shown as a black dashed line. Panel (b) shows ATLAS passive $O_3$ (blue dots), ACE-FTS measurements (green dots), and the ozone loss (red triangles; the difference between observed and average passive ozone) for March 2011.



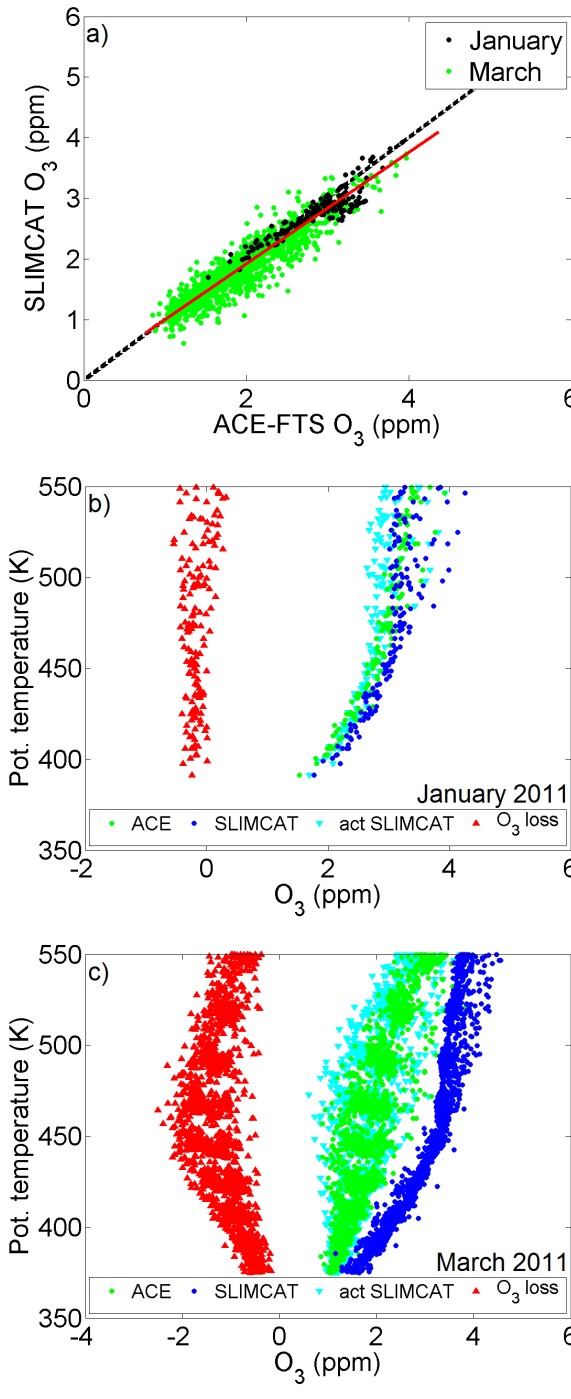

**Figure 5.** Panel (a) shows a comparison between the SLIMCAT ozone and ACE-FTS ozone dataset inside the polar vortex for January (black dots) and March (green dots) 2011, where the regression plot for January and March is shown as a red line. Panel (b) and panel (c) show the comparison between the SLIMCAT ozone (passive ozone shown as blue dots, and ozone as cyan triangles) and measurements (green dots) for January 2011 and March 2011, respectively. The ozone loss is displayed as red triangles and defined as the difference between the measurements and the modelled passive ozone.



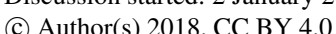

**Figure 6.** The ozone loss estimates are shown using the tracer-tracer correlation technique. Six different tracers have been employed for this method: $CH_4$ (blue), $N_2O$ (light blue), HF (cyan), OCS (yellow), $CCl_2F_2$ (orange), $CCl_3F$ (dark red). The maximum ozone loss profile (in ppmv) is shown in panel (a). For clarity, the uncertainties of the estimated ozone loss profiles have been removed. The integrated ozone loss (in DU) between 380-550 K is shown in panel (b) and (c) for the maximum and mean loss in March, respectively.

**Figure 7.** Same as Fig. 6, but for the average vortex profile descent technique.





**Figure 8.** Similar to Fig. 6, but for the artificial tracer correlation method. Tracer 1 ($N_2O$, $CH_4$, $CCl_2F_2$, $CCl_3F$) in dark blue and Tracer 4 ($N_2O$, $CH_4$, OCS, and $CCl_3F$) in dark red. Details about the composition of the four tracers are provided in the text.





**Figure 9.** Same as Fig. 6, but for the comparison of all different methods: Tracer-tracer correlation (blue), artificial tracer correlation (light blue), average vortex profile descent technique (cyan), passive subtraction with ATLAS (yellow), passive subtraction with SLIMCAT (orange), and passive subtraction using only modelled ozone from SLIMCAT (dark red). An average using different tracers is used for the tracer-tracer correlation, the artificial tracer correlation and the average vortex profile descent technique, see text for details.