# Peer review of "Stratospheric ozone loss in the Arctic winters between 2005 and 2013 derived with ACE-FTS measurements"

_Atmospheric Chemistry and Physics, 2017_

## Referee Comment (RC1) · Anonymous Referee #2 · 20 Jan 2018

**General remarks**

Let me start by saying that this is a very good paper. It has many strengths and using a high quality data set such as ACE-FTS to perform a consistent analysis of Arctic ozone loss constitutes a very valuable scientific study. Using six different methods (and comparing the results) and employing the information from a 'simulation only' analysis is an achievement.

Nonetheless, I have reservations about the paper which concern the conclusions drawn from the paper and the discussion of the different methods. I also think that that the

existing literature on the subject should be discussed in a more balanced way. Further, some additional references (of course not necessarily every single paper mentioned in this review) should be taken into account.

My first general point is that I suggest improving the discussion of the various methods. In the paper it is stated that the tracer-tracer method "neglects descent from high altitudes". This is a bit vague, what means "high altitudes" in this case? Perhaps "above the ozone maximum", where ozone is no longer chemically inert? This should be clarified. And there are a number of studies that address the point of descent from high altitudes in the tracer-tracer method (see below) that should not be neglected. The same is true for the issue of "mixing across the vortex edge" which is discussed for the tracer-tracer method. The basis for the discussion in the manuscript so far is the important work by Michelsen et al. (1998) and Plumb et al. (2000), but there is also a discussion of the arguments presented in these papers in the literature which should not be neglected (e.g., Müller et al., 2005, cited in the manuscript, but not in this context) and (Salawitch et al., 2002, not discussed in the manuscript so far, see also below).

Further, it seems to be tacitly assumed in the manuscript that these issues (high altitude descent and mixing across the edge) do not affect the other methods used in this study. (See for example the discussion on the artificial tracer method below). I strongly suggest to describe the impact of these issues on all considered methods.

Perhaps most importantly, if I accept the conclusion from the paper that the 'descent' method is the most reliable one (at least it seems to be a method with very small error estimates), there appear to be significant differences between the SLIMCAT simulations (SLIMCAT only) and the estimated ozone loss. In the sense that SLIMCAT overestimates the chemical ozone loss. For example for 2009/2010 the simulated loss is $51 \pm 7$ DU and the descent deduced loss is $13 \pm 3$ DU, for the winter 2004/2005, the comparison is $67 \pm 3$ and $47 \pm 4$ DU. (Are these error estimates for ozone loss really comparable). Perhaps I am misreading this conclusion on the reliability of the methods,

but the manuscript should provide a clear guidance to what the message of the paper is in this respect. Thus, I think the paper should discuss these questions in more detail and make a clearer statement on simulated versus observed ozone loss.

In summary, I suggest a more extensive and a more balanced discussion of both the employed methods and the obtained results in a revised version of the manuscript.

**Discussion**

Artificial tracer method

The authors use the artificial tracer method by Esler and Waugh (2002), who have developed the method for $NO_y$ in midlatitudes: they report that they construct an "artificial 'reference tracer' from a linear combination of other long-lived tracers. The reference tracer is designed so that, as far as possible, it has a linear canonical relationship with $NO_y$ in midlatitudes".

The further development of the technique for analysing ozone loss was done by Jin et al. (2006). They state: "The decrease of $O_3$ with respect to this artificial long-lived tracer can be regarded as the chemical $O_3$ loss. However, because the correlations inside and outside the vortex are different, this method cannot correct for the mixing across the vortex edge. This kind of mixing can only increase $O_3$ for an artificial tracer value, which suggests that neglecting mixing across the edge gives a conservative $O_3$ loss estimate for this method". This concept is also illustrated in Fig. 1 of this review.

Thus it is an oversimplification to say that the artificial tracer method compensates for 'mixing'. If mixing occurs across the vortex edge, this will be mixing between two different ozone-tracer relations, even though both could be linear (Fig. 1). And then mixing would have an effect in tracer-tracer space. The authors might not agree with

this concept, but I think a discussion is necessary.

Tracer-tracer method

In the manuscript it is stated that the tracer-tracer method (e.g., p. 7, l. 11) "neglects descent from high altitudes". This is a bit vague, what means "high altitudes" in this case? Perhaps "above the ozone maximum", where ozone is no longer chemically inert? However, e.g. Salawitch et al. (2002) (in a study using tracer-tracer relations) consider data up to 8.9 hPa, which is a relatively high altitude in the stratosphere. They also discuss the question of ozone at higher altitudes (Salawitch et al., 2002, see paragraphs [43] and [44]).

In particular, Salawitch et al. (2002) state that the "*Plumb et al.* [2000] model results for $\chi_2$ versus $\chi_1$ are driven primarily by supply of air at the top of the vortex with near zero mixing ratios of both species. Our observations exhibit a critical difference with respect to these heuristic model calculations. The OMS measurements show that the top of the Arctic vortex is supplied with air having mixing ratios of $O_3$ between 3 and 4 ppm, considerably higher than the final value of [$O_3$] in the inner vortex".

Moreover, there could be intrusions of mesospheric air to lower altitudes, which are discussed by Müller et al. (2007), in a case study. They conclude that "measurements influenced by mesospheric air show ozone mixing ratios ranging between 3.6 and 5.6 ppm, which are clearly greater than those found in the "early vortex" reference relation employed to deduce chemical ozone loss".

Rex et al. (2002) is cited (p. 7., l. 8) in support of the criticism of the tracer-tracer method. And indeed, Rex et al. (2002) mention that "*Michelsen et al.* [1998] and *Plumb et al.* [2000] have suggested that before chemical loss of ozone occurred, mixing between subsided inner vortex air with extravortex air may lead to a flattening out of the curved $O_3/N_2O$ relation and thus may be mistaken as chemical loss of ozone.

However, . . . " But the overall conclusion of Rex et al. (2002) on this issue reads: "Thus the overall changes in the $O_3$ versus $N_2O$ relation observed during the course of winter could not have been caused by transport, and rather represent a lower limit for the true chemical loss of ozone". Therefore, I suggest rethinking of how to use this citation in the paper.

Further, especially for the winter of 1999/2000 there are a number of studies (not taken into account in the manuscript so far) that argue that transport alone could not have led to the observed changes in the $O_3$ versus $N_2O$ relation (tracer-tracer method).

Richard et al. (2001) report that "there is relatively little change in the ER-2 $O_3$:$N_2O$ [. . .] relationships over the two week period between 20 January and 3 February 2000. Additionally, the $O_3$:$N_2O$ profiles are found to be similar to the early winter vortex balloon profiles which allow extension of the relationships to regions above the ER-2 flight altitudes thus defining the chemical composition of air that later descends to ER-2 sampling altitudes (18-21 km). [. . .] Therefore, these relationships allow for the establishment of a winter vortex reference to quantify $O_3$ chemical loss occurring during late February/early March2000."

Another piece of evidence on these issues is provided by Ray et al. (2002) who find that "mixing of midlatitude air into the winter vortex is not a significant contributor to the observed ozone changes in the 1999/2000 season".

Of course, the authors do not need to follow/accept these arguments but I think a more balanced discussion in the manuscript is necessary rather than relying mainly on the arguments of Michelsen et al. (1998) and Plumb et al. (2000) here.

**Comments**

- p. 2., l. 7: "Here we show" – I think it is not really new that these tracers are suitable.

- p.2, l. 20: This is true for the time period of elevated stratospheric chlorine and bromine.

- p. 3., l. 1: Suggest citing here also the early theoretical study by Carslaw et al. (1994).

- p. 3., l. 1, 2: Perhaps helpful: information on observed PSCs is now also available from MIPAS (Spang et al., 2017).

- P. 3., l. 6: The paper by Solomon et al. (1986) is mostly about heterogeneous chlorine activation, less on the relevant catalytic ozone loss cycles.

- p. 3., l. 8: the point is that low temperatures are required but they need to last long enough into the period when sufficient sunlight is available to drive the ozone loss (as it is the case regularly in the Antarctic).

- p.3., l. 28: This statement is also true for the Antarctic. Perhaps look at ozone loss estimates for the Antarctic.

- p. 4., l. 2: I do not think this statement is correct as written here, see the detailed discussion.

- p. 7., l. 8: Rex et al. (2002) is cited here in support of the criticism of the tracer-tracer method. However Rex et al. (2002) state that "Thus the overall changes in the $O_3$ versus $N_2O$ relation observed during the course of winter could not have been caused by transport, and rather represent a lower limit for the true chemical loss of ozone". I think some of the citations used in l. 12 of this page are more appropriate here.

- p.7., l. 12: I cannot see what the contribution of the citation to Michelsen et al. (1998, GRL) is here. This paper does not discuss ozone. Either explain why the citation is needed or drop the citation.

- p. 7., l. 15: add 'over a polar season'; of course tracers like methane or CFC-12 *are* influenced by chemical processes, otherwise there would be no vertical profile.

- p. 7., l. 28: "neglecting mixing processes from the edge [. . . ] over estimation of chemical ozone loss. . . ". I do not think that the papers cited here make this point. See statement from Rex et al. (2002) above. Also, Müller et al. (2005) state in the abstract that "mixing across the polar vortex edge impacts ozone-tracer relations in a way that may solely lead to an 'underestimation' of chemical ozone loss and not to an overestimation"; this discussion needs to be revised.

- p. 10., l. 9: If there is descent from higher altitudes, as discussed in the paper elsewhere; would the 'passive ozone assumption' hold? If ozone is not in complete darkness, it is not passive at higher altitudes.

- p.10., l. 18: what about Wohltmann et al. (2017) here?

- p. 10., l. 32: Why HISPLIT? Would it not be more consistent to calculate the trajectories with the (diabatic) trajectory scheme of ATLAS. And likewise not change the meteorological analysis?

- p. 12., l. 1: Is there a comparison (or comments along this line) between the polar chemistry of SLIMCAT and ATLAS?

- p. 13, l. 14: It is true that it is worrying that tracer profiles do not agree well for March 2005 (which is not shown directly in Fig. 6a). However, I do not understand why this is only a problem for the tracer-tracer method. I suggest that you also discuss the impact of this finding on other ozone loss estimates considered here.

Also the column ozone loss estimates do not seem to differ too much (perhaps with the exception of OCS) for the different tracers in 2005 (Fig. 6).

- p. 13., l. 23: explain why this is likely.

- p. 13., l. 33, 34: it seems obvious to me that average decent profiles will have a small standard deviation, but perhaps I misunderstand. In any event it would be good to give a citation for the smaller uncertainties and how they are calculated.

- p. 18., l. 24: There also could be more comparison here with results in the literature based on the methods used in this study. For example, Tilmes et al. (2006) and Rösevall et al. (2008) report chemical ozone loss for the Arctic winter 2004/2005.

**Details**

- p. 2, l. 32: drop 'ice' here this discussion is about NAT and STS

- p. 3., l. 13: *an* Arctic...

- p. 5., l. 6: trace *gases*

- p. 6., l. 21: I would formulate: . . . "not a sufficient number of measurements" . . .

- p. 8, 9: Eqs. (1) – (4): do not use italics for ppb and ppt

- p. 9., l. 26: here and elsewhere, use proper minus signs; i.e. $-25$ rather than -25

- p. 10., l 10: *is* applied

- p. 11., l. 27: citation for SLIMCAT chemistry scheme?

- p. 19, l. 24: "...ACE, also known as SCISAT" – is this really true?

- p. 19., l. 4: citation for loss in 2010? Also same line 'larger' than what?

- References: there are still a few typos, missing spaces, additional spaces, etc.

**References**

[revised manuscript text omitted]

OUTSIDE

Ozone

INSIDE

Tracer

**Fig. 1.** Schematic view of mixing across the vortex edge for linear ozone tracer relations; see also Fig. 5 in Jin et al. (2006).

---

## Referee Comment (RC2) · Anonymous Referee #3 · 31 Jan 2018

Review of: Stratospheric ozone loss in the Arctic winters between 2005 and 2013 derived with ACE-FTS measurements

By Griffin and colleagues.

General comments

This is a nice paper that does a commendably thorough job of using ozone (and other observations) from the ACE-FTS instrument to quantify, via application of a range of techniques, chemical ozone loss in the Arctic winter/spring. The work is a welcome addition to the field and, in my view, close to being ready for publication in ACP, a journal to which it is well suited. The discussion and results are presented in a very

logical fashion. The standard of English and the quality of the figures etc. are excellent.

I really only have a very few minor comments/suggestions/fixes that should take very little time to implement/explore.

Given that the Match approach has been applied to similar measurements (e.g., from POAM), and one of the authors is highly versed in that technique, it is perhaps a little surprising that that method was not included, or even discussed very much. That said, I can well believe that the ACE-FTS sampling presents a challenge to the implementation of Match-based calculations. Whatever the reason, it would make sense to comment on why it is omitted here. If it's left for "future work", then it's fine to just say that. On the other hand, if there is some reason why it's not practical in this case, it would be good to note it here, as this may prevent others from potentially spending time fruitlessly investigating it in future.

Specific comments

Page 2 line 10: Add a comma after "March 2005" possibly.

Page 7 line 16 and line 20: "blue dots" should be "green dots" in both places. Also, it's a little jarring to be talking about dot color before the figure has been formally introduced (line 16/17).

Page 7 line 29/30: I think "One method that provides a correction for both mixing and for descent..." would be clearer. That is unless I've misunderstood the currently ambiguous wording (it currently could be read as saying that "descent" is another "method" that fixes the mixing issue, rather than another problem to be addressed).

Page 8 line 21: "blue dots" should be "green dots" again.

Page 11 line 5: Just to clarify this is a "horizontal" interpolation only, correct? From the text I get the sense that the vertical "interpolation" is simply "nearest neighbor", correct? Would be good to clarify.

Page 11 line 29: "reset" to what (presumably "ozone that responds to chemistry", but would be good to be clear).

Page 12 line 3: I suggest you change "up to" to "within" and add "great circle" after 0.5<degrees> (unless it's actually latiude or longitude specifically you mean here).

Page 14 lines 14-19: It feels odd to have the "artificial tracer" discussion after the discussion of descent here, given that earlier, in section 3, you introduced those techniques in the other order.

Page 14 line 32: I'd suggest changing "error" to "estimated uncertainties" here, to avoid anyone thinking your taking some kind of inter-method difference as a measure of a (potentially "correctable" error).

Page 15(ish): It does feel a little disjoint to have section 4.1 talking about the various tracer methods, and yet not have any discussion of the ATLAS/SLIMCAT results until you get to the overall intercomparison discussion in 4.2. Might some of the ATLAS/SLIMCAT discussion not merit a subsection of its own.

Page 16 line 7: Here I think you're using "passive subtraction" to only mean the ATLAS/SLMICAT methods, correct? However, in the opening discussion of the manuscript, you have used "passive subtraction" to describe all of your methods (rightly so, as all involve some kind of estimate of passive ozone). Might be better to use a different term here.

Figure 1. I'm curious as to where the cluster of black points ("fliers" actually) with O3 around 4.5-5 ppmv in panels a,b,c and d have "gone" in e and f? Are these cases where there were no OCS or CCl3F measurements? Or are they all hiding under the "e)" and "f)" legends (I hope not). Also, in the former cases (a-d) I would expect that they may be contributing significantly to the "uncertainty" in the fit. Might there be something geophysically unusual about them (their ozone abundance clearly implies as much) that would give you a good basis for discounting them? Also, you might want

to think about moving the a-f legends to a different corner of the plot to avoid clutter.

Figure 2: I'd move the legend (January, March) somewhere else so it doesn't get in the way. Also you don't need it on all four panels (you only had it on one panel in figure 1). That should make it easier for you to find an out of the way place.

Figure 5, caption, line 2: "...2011, with the combined regression fit for January and March...", assuming that's a correct interpretation.
* * *

---

## Referee Comment (RC3) · B.-M. Sinnhuber (Referee) · 26 Feb 2018

In this study, Griffin et al. estimate Arctic stratospheric ozone loss from ACE-FTS measurements using a range of different techniques. This is an important study addressing the consistency of ozone loss estimates using different techniques and their inherent assumptions. The paper is well written and merits publication in Atmos. Chem. Phys. after addressing of the following comments.

(I would like to apologize for my late review and have no other excuse than juggling with too many things at the same time.)

[Figure]

General comments:

1) "artificial tracer": The argument that mixing does not change the correlation between ozone and the artificial tracer (p4, l4) is only true if the correlation exhibits the same slopes inside and outside the polar vortex. If not, than mixing across the vortex edge can influence the correlation. On p7, l30 it is stated that this method provides a "mixing correction". This is not immediately clear. As this is a critical point, I suggest to show the correlations inside and outside of the vortex.

2) Uncertainty of passive subtraction with ATLAS (p11, l8): Estimating the uncertainty by comparing ATLAS and ACE-FTS for January will almost certainly underestimate the true uncertainty, as the model was initialized in early January and run only for a relatively short period – uncertainties in model transport will accumulate until March, not captured here. While it is difficult to come up with a better uncertainty estimate, this needs to be at least acknowledged and discussed.

Specific comments and technical corrections:

P3, l13: "Polar Night Jet Oscillation Event": I suggest either to give more information or drop the reference to the Polar Night Jet Oscillation Event. What is this and why is this relevant?

P4, l13: estimate differences between model and observations: The meaning of this sentence is not fully clear and should be rephrased accordingly.

P4, l20: "...and the passive subtraction method using only modelled ozone": If the meaning here is "...and compare this to the modelled chemical ozone loss" better say so.

P7,l11: "high altitudes": upper stratosphere and mesosphere?

P13, l11: "uncertainty 10-20%": absolute or 10-20% of the ozone loss?

P13, l13: "that further confirmed the tracer/tracer correlation method to be inaccurate

for estimating Arctic ozone loss": This is a strong statement. Do you really want to say tracer/tracer methods are inaccurate for ozone loss estimates?

P14, l24: does this apply specifically to ACE-FTS retrievals of OCS and CCl3F ? If so it would be good to mention explicitly.

P16, l6: "The results of the artificial tracer technique should be uninfluenced by mixing": Again, it needs to be demonstrated that this is also true for mixing across the vortex edge.

P16, l9: "The passive subtraction methods may smooth out the year-to-year differences and model results in some years may compute some ozone loss even in the absence of ClOx chemistry": Why?? The meaning and basis for this statement is unclear.

P16, l22: "ozone loss has also been estimated using only the SLIMCAT ozone and passive ozone ("SLIMCAT only")": Again, I believe this is better expressed as "modeled ozone loss".

P18, l33: "passive subtraction methods using either ATLAS or SLIMCAT seem to have smaller computed uncertainties": As remarked above, I suspect that for these methods the uncertainties here are systematically underestimated.

P19, l4: "and might smooth out the year-to-year variability": again, any idea why the year-to-year variability may be "smoothed out"?

P19, l11: "For years with little to no ClOx activation the artificial tracer correlation technique might be the most reliable because it considers mixing and seems to compute a reasonably small ozone loss": This statement is problematic for two reasons: (a) one may argue that possible mixing across the vortex edge is better represented by the passive subtraction method that takes into account tracer gradients across the vortex edge at least in first order, and (b) the relatively good agreement between the passive subtraction method and modeled ozone loss ("SLIMCAT only") for this year (2010) indicates that according to our understanding of the processes involved there was potential

for substantial chemical loss.

---

## Author Comment (AC1) · 9 Jul 2018

We would like to thank reviewer #2 for his/her corrections and recommendations. Additions to the text are highlighted in blue and text that has been removed from the original text is highlighted in red. The reviewer comments are included in bold.

**General remarks**

**Let me start by saying that this is a very good paper. It has many strengths and using a high quality data set such as ACE-FTS to perform a consistent analysis of Arctic ozone loss constitutes a very valuable scientific study. Using six different methods (and comparing the results) and employing the information from a 'simulation only' analysis is an achievement.**

**Nonetheless, I have reservations about the paper which concern the conclusions drawn from the paper and the discussion of the different methods. I also think that that the existing literature on the subject should be discussed in a more balanced way. Further, some additional references (of course not necessarily every single paper mentioned in this review) should be taken into account.**

**My first general point is that I suggest improving the discussion of the various methods. In the paper it is stated that the tracer-tracer method "neglects descent from high altitudes". This is a bit vague, what means "high altitudes" in this case? Perhaps "above the ozone maximum", where ozone is no longer chemically inert? This should be clarified. And there are a number of studies that address the point of descent from high altitudes in the tracer-tracer method (see below) that should not be neglected. The same is true for the issue of "mixing across the vortex edge" which is discussed for the tracer-tracer method. The basis for the discussion in the manuscript so far is the important work by Michelsen et al. (1998) and Plumb et al. (2000), but there is also a discussion of the arguments presented in these papers in the literature which should not be neglected (e.g., Mller et al., 2005, cited in the manuscript, but not in this context) and (Salawitch et al., 2002, not discussed in the manuscript so far, see also below).**

**Further, it seems to be tacitly assumed in the manuscript that these issues (high altitude descent and mixing across the edge) do not affect the other methods used in this study. (See for example the discussion on the artificial tracer method below). I strongly suggest to describe the impact of these issues on all considered methods.**

**Perhaps most importantly, if I accept the conclusion from the paper that the descent method is the most reliable one (at least it seems to be a method with very small error estimates), there appear to be significant differences between the SLIMCAT simulations (SLIMCAT only) and the estimated ozone loss. In the sense that SLIMCAT overestimates the chemical ozone loss. For example for 2009/2010 the**

simulated loss is 51 - 7 DU and the descent deduced loss is 13 $\pm 3$ DU, for the winter 2004/2005, the comparison is $67 \pm 3$ and $47 \pm 4$ DU. (Are these error estimates for ozone loss really comparable). Perhaps I am misreading this conclusion on the reliability of the methods, but the manuscript should provide a clear guidance to what the message of the paper is in this respect. Thus, I think the paper should discuss these questions in more detail and make a clearer statement on simulated versus observed ozone loss. In summary, I suggest a more extensive and a more balanced discussion of both the employed methods and the obtained results in a revised version of the manuscript.

We thank the reviewer for the thorough review and great suggestions to improve our paper. These general comments have been addressed and detailed comments are provided in the revisions below.

Artificial tracer method
The authors use the artificial tracer method by Esler and Waugh (2002), who have developed the method for $NO_y$ in midlatitudes: they report that they construct an "artificial 'reference tracer' from a linear combination of other long-lived tracers. The reference tracer is designed so that, as far as possible, it has a linear canonical relationship with $NO_y$ in midlatitudes".
The further development of the technique for analysing ozone loss was done by Jin et al. (2006). They state: "The decrease of $O_3$ with respect to this artificial long-lived tracer can be regarded as the chemical $O_3$ loss. However, because the correlations inside and outside the vortex are different, this method cannot correct for the mixing across the vortex edge. This kind of mixing can only increase $O_3$ for an artificial tracer value, which suggests that neglecting mixing across the edge gives a conservative $O_3$ loss estimate for this method". This concept is also illustrated in Fig. 1 of this review.
Thus it is an oversimplification to say that the artificial tracer method compensates for 'mixing'. If mixing occurs across the vortex edge, this will be mixing between two different ozone-tracer relations, even though both could be linear (Fig. 1). And then mixing would have an effect in tracer-tracer space. The authors might not agree with this concept, but I think a discussion is necessary.

We have included further discussion of the artificial tracer method, correcting the statement that the artificial tracer correlation method corrects for mixing across the vortex edge. The first paragraph of this Sect. 3.2 has been changed according to the suggestions provided by the reviewer:

"The amount of mixing of extra-vortex air into the polar vortex varies widely depending on the dynamics of each winter and spring, and is more likely to occur in the NH (WMO, 2014). Neglecting mixing processes from the edge of

the polar vortex  or mixing of high altitude air (above the ozone maximum) can lead to an underestimation of the chemical ozone loss (e.g., Rex et al., 2002; Müller et al., 2005). One method that provides a , in addition to correction for both mixing from the vortex edge and for descent, is the artificial tracer method. This method was first proposed by Esler and Waugh (2002) and uses a "tracer" created from a linear combination of several different trace gases that is linearly correlated with ozone. This linear correlation makes it easier to determine the ozone loss and reduces the impact of mixing, since mixing from the edge of the vortex would only result in "moving" the air parcels along this linear correlation line (Esler and Waugh,2002). Initially such an artificial tracer method was used by Esler and Waugh (2002) to estimate denitrification inside the Arctic polar vortex. However, this same method can be applied to estimating the chemical ozone loss as was done by Jin et al. (2006). While it reduces the error from mixing of air near the vortex edge, this method, however, does not account for mixing of extra-polar vortex air into the vortex. The artificial tracer, established from observations inside the polar vortex does not follow the same linear correlation outside the polar vortex (Jin et al., 2006).

We also changed the fourth paragraph in Sect. 4.2, p.16, l.15:

"Discrepancies are apparent between the measurement only methods and the passive subtraction  using CTMs for 2010, especially for the computed mean partial column loss. Each time the vortex splits and the two parts reunite, extra-vortex air is mixed. In 2010 the polar vortex was very disturbed, therefore,  methods that do not account for the mixing of extra-vortex air (the tracer-tracer  method, the profile descent techniques and the artificial tracer technique) are not reliable for that year since an isolated vortex is essential for these methods. The loss estimates in 2010 using the  measurement only techniques do not agree with the passive subtraction  using CTMs. Generally, we see the largest differences between the passive subtraction method using CTMs and methods that use measurements only for years with strong turbulence and relatively small ozone loss (see Table 1).For example in 2010,  the passive subtraction methods using CTMs are nearly twice as high for the maximum ozone loss and more than three times as high for the mean ozone loss than the methods that use measurements only. This could either be due to mixing processes unaccounted for in the methods using measurements only or the passive subtraction methods using CTMs may overestimate passive ozone."

Additionally, we have changed our wording in the final conclusions and discussion:

"  Based on this study, for years with a stable and strong polar vortex, the tracer-tracer technique, the artificial tracer technique and the passive subtraction using both CTMs lead to similar ozone losses and seem to estimate a similar passive ozone profile. We also found that from the six different estimation methods presented,  the artificial tracer correlation technique  and the passive subtraction method (with ATLAS or SLIMCAT)  are best suited for estimating the ozone loss in the Arctic polar vortex.  For years with an unstable polar vortex we recommend using the passive subtraction  technique, since the artificial tracer  technique does not account for mixing of extra-polar vortex air. We did not find any difference between an Eulerian or a Lagrangian model and found that both types of CTMs seem to compute the Arctic ozone loss equally well. . "

**Tracer-tracer method**
In the manuscript it is stated that the tracer-tracer method (e.g., p. 7, l. 11) "neglects descent from high altitudes". This is a bit vague, what means "high altitudes" in this case? Perhaps "above the ozone maximum", where ozone is no longer chemically inert? However, e.g. Salawitch et al. (2002) (in a study using tracer-tracer relations) consider data up to 8.9 hPa, which is a relatively high altitude in the stratosphere. They also discuss the question of ozone at higher altitudes (Salawitch et al., 2002, see paragraphs [43] and [44]).
In particular, Salawitch et al. (2002) state that the "*Plumb et al. [2000]* model results for $\chi_2$ versus $\chi_1$ are driven primarily by supply of air at the top of the vortex with near zero mixing ratios of both species. Our observations exhibit a critical difference with respect to these heuristic model calculations. The OMS measurements show that the top of the Arctic vortex is supplied with air having mixing ratios of $O_3$ between 3 and 4 ppm, considerably higher than the final value of $[O_3]$ in the inner vortex". Moreover, there could be intrusions of mesospheric air to lower altitudes, which are discussed by Müller et al. (2007), in a case study. They conclude that "measurements influenced by mesospheric air show ozone mixing ratios ranging between 3.6 and 5.6 ppm, which are clearly greater than those found in the "early vortex" reference relation employed to deduce chemical ozone loss".
Rex et al. (2002) is cited (p. 7., l. 8) in support of the criticism of the tracer-tracer method. And indeed, Rex et al. (2002) mention that "*Michelsen et al. [1998]* **and** *Plumb et al. [2000]* **have suggested that**

before chemical loss of ozone occurred, mixing between subsided inner vortex air with extravortex air may lead to a flattening out of the curved $O_3/N_2O$ relation and thus may be mistaken as chemical loss of ozone. However, ... " But the overall conclusion of Rex et al. (2002) on this issue reads: "Thus the overall changes in the $O_3$ versus $N_2O$ relation observed during the course of winter could not have been caused by transport, and rather represent a lower limit for the true chemical loss of ozone". Therefore, I suggest rethinking of how to use this citation in the paper.

Further, especially for the winter of 1999/2000 there are a number of studies (not taken into account in the manuscript so far) that argue that transport alone could not have led to the observed changes in the O3 versus N2O relation (tracer-tracer method).

Richard et al. (2001) report that "there is relatively little change in the ER-2 $O_3$:$N_2O$ [... ] relationships over the two week period between 20 January and 3 February 2000. Additionally, the $O_3$:$N_2O$ profiles are found to be similar to the early winter vortex balloon profiles which allow extension of the relationships to regions above the ER-2 flight altitudes thus defining the chemical composition of air that later descends to ER-2 sampling altitudes (18-21 km). [... ] Therefore, these relationships allow for the establishment of a winter vortex reference to quantify O3 chemical loss occurring during late February/early March2000."

Another piece of evidence on these issues is provided by Ray et al. (2002) who find that "mixing of midlatitude air into the winter vortex is not a significant contributor to the observed ozone changes in the 1999/2000 season".

Of course, the authors do not need to follow/accept these arguments but I think a more balanced discussion in the manuscript is necessary rather than relying mainly on the arguments of Michelsen et al. (1998) and Plumb et al. (2000) here.

We have re-written some of the text in the paper and included a more balanced discussion of the tracer-tracer correlation method as suggested by the reviewer, and included the references as suggested.
We have changed the first paragraph of Sect. 3.1 according to the reviewer's suggestions:

"As described in Sect. 2.2, measurements taken in January inside the polar vortex are used to quantify the ozone distribution before significant ozone depletion occurs. This dataset is then compared to measurements taken in March, when chemical ozone depletion is most pronounced in the observed ozone profile. This method has been criticized for neglecting processes that mix extra-vortex air into the polar vortex (e.g., Rex et al.,2002) (e.g., Michelsen et al., 1998b; Plumb et al., 2000, 2003; Plumb, 2007), because it assumes that the polar vortex is isolated, which is not true for all years, especially in the Arctic. By On the

other hand, some studies observing Arctic ozone loss the 1999/2000 winter (a winter with an unusually strong polar vortex and thus little mixing) have found that mixing of mid-latitude air was not a significant contributor to the observed changes (e.g., Richard et al., 2001; Ray et al., 2002). In our study, using the sPV criteria described above, we attempt to limit the influence of mixing of extra-vortex air in our calculation of the early vortex reference function.

The tracer-tracer correlation method also neglects descent of ozone or the tracer from high altitudes  (middle and upper stratosphere and mesosphere) above 550 K that is not included in our calculation of the early vortex reference function. However, Salawitch et al. (2002) showed that supply of ozone depleted air into the top of the vortex did not play a role in the subsequent evolution of the ozone-tracer relation in the 1999/2000 Arctic winter (where the vortex was strong). Mixing of air from top of the Arctic vortex (where mixing ratios are between 3 and 4 ppm (Salawitch et al., 2002)) into the polar vortex, could, however, underestimate the ozone loss of the tracer-tracer method. Rex et al. (2002) state that the tracer-tracer correlation represents a lower limit of the true ozone loss in the case of the 1999/2000 Arctic winter ( a year with a stable polar vortex). "

And we included changes in Sect. 4.2, p.16, l.1:

"In some years, the tracer-tracer correlation method and the average vortex descent technique differ significantly from all other estimation methods. These  differences highlight the difficulty of using the tracer-tracer correlation method, because mixing processes and descent in the 2005 Arctic vortex are not  accounted for. These differences also highlight the difficulty of using the average vortex descent technique in years of an unstable polar vortex.  The average polar vortex descent technique  typically underestimates the ozone loss compared to all other methods, this technique only agrees well with the other methods in March 2007  and 2008."

**Comments**

**p. 2., l. 7: "Here we show" I think it is not really new that these tracers are suitable.**

We changed the sentence to:

"For the tracer-tracer, the artificial tracer, and the average vortex profile descent

approaches, various tracers have been used  that are measured by ACE-FTS. From these seven tracers investigated, we found that $CH_4$, $N_2O$, HF, and CFC-12 are the most suitable tracers for investigating polar stratospheric ozone depletion with ACE-FTS."

**p.2, l. 20: This is true for the time period of elevated stratospheric chlorine and bromine.**

We have changed the sentence to:

"Arctic ozone column loss is extremely variable in the winter/springtime and can range from near zero to about 150 DU..."

**p. 3., l. 1: Suggest citing here also the early theoretical study by Carslaw et al. (1994).**

The citation has been included.

**p. 3., l. 1, 2: Perhaps helpful: information on observed PSCs is now also available from MIPAS (Spang et al., 2017).**

The citation has been included.

**P. 3., l. 6: The paper by Solomon et al. (1986) is mostly about heterogeneous chlorine activation, less on the relevant catalytic ozone loss cycles.**

The citation has been removed, only McElroy et al. (1986) and Molina and Molina (1987) are cited in this sentence now.

**p. 3., l. 8: the point is that low temperatures are required but they need to last long enough into the period when sufficient sunlight is available to drive the ozone loss (as it is the case regularly in the Antarctic).**

This sentence has been added to clarify:

"For polar ozone loss, low temperatures are required but they also need to last long enough into the period when sufficient sunlight is available to drive the ozone loss.

**p.3., l. 28: This statement is also true for the Antarctic. Perhaps look at ozone loss estimates for the Antarctic.**

We wanted to highlight that the dynamic variability is stronger in the Arctic, and SSW occur more frequently in the Northern Hemisphere. The sentence

has been rephrased to:

"Because of the  dynamical variability of the Arctic polar vortex, quantifying chemical ozone loss  in the Arctic."

**p. 4., l. 2: I do not think this statement is correct as written here, see the detailed discussion.**

The sentence has been changed to:

"Using an artificial tracer(e.g., Esler and Waugh, 2002; Jin et al., 2006) that is constructed (from observed trace gases) to be linearly correlated with ozone can improve the accuracy of the loss estimate."

**p. 7., l. 8: Rex et al. (2002) is cited here in support of the criticism of the tracer-tracer method. However Rex et al. (2002) state that "Thus the overall changes in the $O_3$ versus $N_2O$ relation observed during the course of winter could not have been caused by transport, and rather represent a lower limit for the true chemical loss of ozone". I think some of the citations used in l. 12 of this page are more appropriate here.**

As described above (in the discussion of the artificial tracer technique), we have included these suggested changes in this paragraph.

**p.7., l. 12: I cannot see what the contribution of the citation to Michelsen et al. (1998, GRL) is here. This paper does not discuss ozone. Either explain why the citation is needed or drop the citation.**

Michelsen et al. (1998,GRL) is a critical reference showing observational evidence for different mixing lines inside and outside the vortex and at different times within the vortex, and as such is a foundation for any use of tracer correlation methods for ozone loss.

**p. 7., l. 15: add 'over a polar season'; of course tracers like methane or CFC- 12 *are* influenced by chemical processes, otherwise there would be no vertical profile.**

This has been added as suggested.

"A tracer is required to be long-lived and stable (Plumb and Ko, 1992) and thus, not influenced by chemical processes over a polar season."

**p. 7., l. 28: "neglecting mixing processes from the edge [... ] over estimation of chemical ozone loss... ". I do not think that the papers**

**cited here make this point. See statement from Rex et al. (2002) above. Also, Müller et al. (2005) state in the abstract that "mixing across the polar vortex edge impacts ozone-tracer relations in a way that may solely lead to an 'underestimation' of chemical ozone loss and not to an overestimation"'; this discussion needs to be revised.**

We have revised this discussion accordingly, as stated above (in the discussion of the tracer-tracer technique).

**p. 10., l. 9: If there is descent from higher altitudes, as discussed in the paper elsewhere; would the 'passive ozone assumption' hold? If ozone is not in complete darkness, it is not passive at higher altitudes.**

In this study we focus on $O_3$ between approximately 380 to 550 K. For this altitude (the lower stratosphere), the time-scale for dynamical changes is much shorter that that for chemical changes (unless there is heterogeneous PSC-mediated chemistry). The region where chemical and dynamical time-scales are similar is around 30 to 40 km (800-1000K). The region studied in this paper is well below that altitude, and below the region where gas-phase chemistry would affect the passive ozone in situ. If ozone was transported down from 700 or 800K to the 350-550 K region, in its new environment any chemical reactions would take place at the rates consistent with that altitude, and thus would be very slow compared to those for transport. So any significant affect on the passive ozone should be negligible at and below 550 K. E.g., Singleton et al. (2005) compared two model runs with the gas phase chemistry turned on and the other one off, the differences increased with altitude, but remained below 0.5 ppmv for 550 K and are negligible for lower altitudes in March.

Singleton, C. S., Randall, C. E., Chipperfield, M. P., Davies, S., Feng, W., Bevilacqua, R. M., Hoppel, K. W., Fromm, M. D., Manney, G. L., and Harvey, V. L.: 2002-2003 Arctic ozone loss deduced from POAM III satellite observations and the SLIMCAT chemical transport model, Atmos. Chem. Phys., 5, 597-609, https://doi.org/10.5194/acp-5-597-2005, 2005.

**p.10., l. 18: what about Wohltmann et al. (2017) here?**

We have included the citation.

**p. 10., l. 32: Why HISPLIT? Would it not be more consistent to calculate the trajectories with the (diabatic) trajectory scheme of ATLAS. And likewise not change the meteorological analysis?**

For practical purposes the trajectories were estimated with HYSPLIT. We agree that possibly it would have been more consistent to calculate trajectories with the diabatic trajectory scheme of ATLAS, however, the changing this would require and effort that is not justified by the benefit. Small changes of the trajectories will have a negligible effect on the end results.

**p. 12., l. 1: Is there a comparison (or comments along this line) between the polar chemistry of SLIMCAT and ATLAS?**

In this study, we did not use the results from the chemical model of ATLAS, and therefore did not discuss the differences between the chemical ozone scheme of the ALTAS and SLIMCAT model. As such, we did not perform a detailed comparison between SLIMCAT and ATLAS, and there is currently no paper on this subject.

**p. 13, l. 14: It is true that it is worrying that tracer profiles do not agree well for March 2005 (which is not shown directly in Fig. 6a). However, I do not understand why this is only a problem for the tracer-tracer method. I suggest that you also discuss the impact of this finding on other ozone loss estimates considered here. Also the column ozone loss estimates do not seem to differ too much (perhaps with the exception of OCS) for the different tracers in 2005 (Fig. 6).**

We have removed the last two sentences of this paragraph and included the following sentence on p.13:

"This indicates the  shortcomings of the tracer-tracer correlation method, even  in case where only inner core vortex measurements were used for estimating the ozone loss. These results are  consistent with previous studies (e.g., Michelsen et al., 1998 a, b, 2000; Plumb et al., 2000, 2003; Plumb, 2007) that  have shown tracer-tracer  correlations are not expected to be accurate for estimating Arctic ozone loss. However, in this study, though the profile loss estimates are different for different tracers, the partial column losses (maximum and mean) are not significantly different and agree within the estimated uncertainties. "

We have also added a sentence describing the profiles for the other two methods, on p. 14:
"The profile loss estimated for these different tracers looks similar for most years with the exemption of OCS in 2005, 2008 and 2011, and $CCl_3F$ in 2010."

**p. 13., l. 23: explain why this is likely.**

We have changed the sentence to:

"Also, in 2007, the estimated loss is larger  when HF is used as a tracer,  and does not follow the ozone loss profile as estimated with other tracers."

**p. 13., l. 33, 34: it seems obvious to me that average decent profiles will have a small standard deviation, but perhaps I misunderstand. In any event it would be good to give a citation for the smaller uncertainties and how they are calculated.**

Section 3.2 explains how the uncertainties are estimated for the average vortex profile descent technique and highlights some of the difficulties estimating this uncertainty that might result in underestimating the true uncertainty. The following has been added:

"Note, this method only allows the estimation of one vortex averaged passive ozone profile; all other methods applied in this study estimate a passive ozone mixing ratio for each data point in March. Consequently, this method does not consider any changes of the passive ozone levels that can occur throughout March. The uncertainty of the passive ozone is estimated based on the $\pm 1\sigma$ standard deviation of the average vortex profile descent (that is quite small for the average vortex profile descent technique). To obtain the total uncertainty, the statistical fitting error of the ACE-FTS tracer measurements and the uncertainty of the passive ozone are added in quadrature. This uncertainty estimate is based on statistical errors only and as such underestimates the true uncertainty. It is difficult to estimate the true uncertainty in this case, because of the unknown effect of ozone due to mixing processes."

And we have included the following sentence in Sect. 4.1, p.14, l.12:

"Note that this does not represent the true uncertainty but more of a statistical uncertainty, since there is only one passive ozone profile for each March (and, therefore, the same amount of ozone at each potential temperature level) the uncertainty is likely much higher. "

Additionally, we have included a sentence highlighting some of the difficulties using the average polar vortex descent technique, in last paragraph of Sect. 4.2, p. 17 l. 26:

"Overall, we have found that the different methods agree in most years within the estimated uncertainties considering the profile mixing ratio loss, as well as the mean and maximum partial column ozone loss. Typically, the average vortex profile descent method estimates smaller ozone losses compared to all other methods. This method provides an approximate ozone loss estimate, however, from only one passive ozone profile, and hence, the passive ozone is the same throughout the month at each potential temperature level"

**p. 18., l. 24: There also could be more comparison here with results in the literature based on the methods used in this study. For example, Tilmes et al. (2006) and Rösevall et al. (2008) report chemical ozone loss for the Arctic winter 2004/2005.**

We have added the following in Sect. 4.2:

"A comparable partial column loss (120 DU) to our loss estimate using the tracer-tracer correlation method has been estimated by Tilmes et al., 2006 with satellite-borne HALOE observations using the tracer-tracer correlation method.The peak ozone loss in 2005 has also been estimated by Rösevall et al. (2008) using the tracer-tracer correlation technique (with the satellite-borne MLS and Sub-Millimetre Radiometer (SMR) instruments) that is around 1 ppmv and more comparable with our other loss estimates. "

**p. 2, l. 32: drop 'ice' here this discussion is about NAT and STS**

We have changed the sentence accordingly to:

" PSCs that contain primarily  particles..."

**p. 3., l. 13: an Arctic...**

This sentence has been modified according to Björn-Martin Sinnhuber's review and this part of the sentence has been removed.

"In January 2012, very strong polar vortex disturbance occurred  (Berhard et al., 2012; Chandran et al., 2013) ."

**p. 5., l. 6: trace gases**

This has been fixed.

**p. 6., l. 21: I would formulate: ... "not a sufficient number of measurements" ...**

The sentence has been changed to:

"... consequently there were not sufficient number of measurements inside the polar vortex in March to perform the analysis with ACE-FTS."

**p. 8, 9: Eqs. (1) (4): do not use italics for ppb and ppt**

This has been fixed.

**p. 9., l. 26: here and elsewhere, use proper minus signs; i.e. $-25$ rather than -25**

We changed the minus signs accordingly throughout the text.

**p. 10., l 10:** *is* **applied**

We changed this sentence accordingly.

**p. 11., l. 27: citation for SLIMCAT chemistry scheme?**

We changed the sentence to:

"It contains a detailed stratospheric chemistry scheme including all processes that are related to polar ozone depletion (Chipperfield et al., 2006; Dhomse et al., 2013; and references therein). ."

**p. 19, l. 24: "...ACE, also known as SCISAT" is this really true?**

The CSA refers to ACE in this manner.

**p. 19., l. 4: citation for loss in 2010? Also same line 'larger' than what?**

We have changed the sentence to:

"For a highly disturbed vortex, e.g. , the passive subtraction methods using CTMs indicate larger ozone loss  than the methods that use measurements only, indicating that either measurement only methods underestimate the ozone loss due to unaccounted mixing processes or the the passive subtraction methods using CTMs might smooth out the year-to-year variability by overestimating passive ozone. "

**References: there are still a few typos, missing spaces, additional spaces, etc.**

We have corrected the following typos and spelling mistakes that we were able to find. Please let us know if there are any more specific typos and spelling errors that we should correct in the list of references.

Bernhard, G., Manney, G., Fioletov, V., Groo, J.-U., Heikkilä, A., Johnsen, B., Koskela, T., Lakkala, K., Müller, R.,  Myhre, C. L., and Rex, M.: Ozone and UV Radiation, in: State of the Climate 2011, Bull. Amer. Meteor. Soc., 93 (7), S129–S132, 2012.

Fromm, M. D., Bevilacqua, R. M., Hornstein, J., Shettle, E. P., Hoppel, K., and Lumpe, J. D.: An analysis of Polar Ozone and Aerosol Measurement POAM II Arctic stratospheric cloud observations, 1993–1996, J. Geophys. Res., 104, 24,34124,357, 1999.

Goutail, F., Pommereau, J.-P., Lefèvre, F., van Roozendael, M., Andersen, S. B., Kåstad Høiskar, B.-A., Dorokhov, V., Kyrö, E., Chipperfield, M. P., and Feng, W.: Early unusual ozone loss during the Arctic winter 2002/2003 compared to other winters, Atmos. Chem. Phys, 5, 665–677, doi:10.5194/acp-5-665-2005, 2005.

Harris, N. R. P., Rex, M., Goutail, F., Knudsen, B. M., Manney, G. L., Müller, R. and von der Gathen , P.: Comparison of empirically derived ozone loss rates in the Arctic vortex, J. Geophys. Res., 107, D20, SOL 7-1–SOL 7-11, doi:10.1029/2001JD000482, 2002.

Hoffmann, L., Hoppe, C. M., Müller, R., Dutton, G. S., Gille, J. C., Griessbach, S., Jones, A., Meyer, C. I., Spang, R., Volk, C. M., and Walker, K. A.: Stratospheric lifetime ratio of CFC-11 and CFC-12 from satellite and model climatologies, Atmos. Chem. Phys., 14, 12479-12497, doi:10.5194/acp-14-12479-2014, 2014.

Lowe, D., and MacKenzie, A. R.: Polar stratospheric cloud microphysics and chemistry, J. Atmos. Sol.-Terr. Phys., 70, 13–40, 2008.

Rex, M., von der Gathen, P., Harris, N. R. P., Lucic, D., Knudsen, B. M., Braathen, G. O., Reid, S. J., De Backer, H., Claude, H., Fabian R., Fast, H., Gil, M., Kyrö, E., Mikkelsen, I. S., Rummukainen, M., Smit, H. G., Stähelin, J., Varotsos, C., and Zaitcev, I.: In situ measurements of stratospheric ozone depletion rates in the Arctic winter 1991/1992: A Lagrangian approach, J. Geophys. Res., 103, D5, 5843–5853, 1998.

Rienecker, M. M., Suarez, M. J., Todling, R., Bacmeister, J., Takacs, L., Liu, H.-C., Gu, W., Sienkiewicz, M., Koster, R. D., Gelaro, R., Stajner, I., and Nielsen, J. E.: The GEOS-5 data assimilation system – documentation of versions 5.0.1, 5.1.0, and 5.2.0, NASA Tech. Memo., TM-2008-104606, 27, 2008.

---

## Author Comment (AC2) · 9 Jul 2018

We would like to thank reviewer #3 for his/her corrections and recommendations. Additions to the text are highlighted in blue and text that has been removed from the original text is highlighted in red. The reviewer comments are included in bold.

**Given that the Match approach has been applied to similar measurements (e.g., from POAM), and one of the authors is highly versed in that technique, it is perhaps a little surprising that that method was not included, or even discussed very much. That said, I can well believe that the ACE-FTS sampling presents a challenge to the implementation of Match-based calculations. Whatever the reason, it would make sense to comment on why it is omitted here. If its left for "future work", then its fine to just say that. On the other hand, if there is some reason why its not practical in this case, it would be good to note it here, as this may prevent others from potentially spending time fruitlessly investigating it in future.**

The reason for not including the Match approach is that due to the orbit of ACE-FTS there is a measurement gap in the Arctic in February that is typically 2-3 weeks. This time period is too long for trajectory estimations to match the observations and track the air parcels. We have investigated applying the Match approach, but it would only be possible to use this approach either in January or March, but not over the entire winter/spring period that was investigated in this study.

**Specific comments:**

**Page 2 line 10: Add a comma after "March 2005" possibly.**

We have changed the text accordingly.

**Page 7 line 16 and line 20: "blue dots" should be "green dots" in both places. Also, its a little jarring to be talking about dot color before the figure has been formally introduced (line 16/17).**

The sentence has been changed to:

" Figure 1 shows the $O_3$-tracer correlation between for these six tracers for the winter/spring 2011, displayed are the ACE-FTS measurements in January (black dots) and March ( green dots) together with the estimated early vortex reference function (red solid line) . "

**Page 7 line 29/30: I think "One method that provides a correction for both mixing and for descent..." would be clearer. That is unless Ive misunderstood the currently ambiguous wording (it currently could be read as saying that "descent" is another "method" that fixes the**

**mixing issue, rather than another problem to be addressed).**

The sentence has been changed accordingly to:

"One method that provides a  correction for both mixing from the vortex edge and for descent, is the artificial tracer method."

**Page 8 line 21: "blue dots" should be "green dots" again.**

This has been corrected.

**Page 11 line 5: Just to clarify this is a "horizontal" interpolation only, correct? From the text I get the sense that the vertical "interpolation" is simply "nearest neighbor", correct? Would be good to clarify.**

Yes, the interpolation is only horizontal. But for the vertical, we simply used the points that have the same potential temperature levels as the ACE-FTS measurements within the the ATLAS resolution without interpolation. To clarify, we added the following sentence in Sect. 3.4.1 (p.11):
"The interpolation is only done horizontally, we did not apply interpolation in the vertical direction but instead chose only ATLAS points that were at the same potential temperature levels as the ACE-FTS observations, within the resolution of ATLAS. "

**Page 11 line 29: "reset" to what (presumably "ozone that responds to chemistry", but would be good to be clear).**

We have changed the sentence to be more clear:

"The passive ozone from the SLIMCAT model run was reset on 1 January for each year to the values of the model chemical ozone field at that time."

**Page 12 line 3: I suggest you change "up to" to "within" and add "great circle" after 0.5 degrees (unless its actually latitude or longitude specifically you mean here).**

We changed the sentence accordingly:

"Although the geo-location of the ACE-FTS measurements change with altitude, the location of the measurements at the altitudes of interest (approximately 15-25 km) are  within an approximately 0.5° great circle of the location of the 30 km tangent altitude and, therefore, within the model resolution."

**Page 14 lines 14-19: It feels odd to have the "artificial tracer" discussion after the discussion of descent here, given that earlier, in section 3, you introduced those techniques in the other order.**

The order in section 3 has been changed to 3.1 Tracer-tracer method, 3.2 Average vortex profile descent technique, 3.3 Artificial tracer method.

**Page 14 line 32: Id suggest changing "error" to "estimated uncertainties" here, to avoid anyone thinking your taking some kind of inter-method difference as a measure of a (potentially "correctable" error).**

This has been changed accordingly.

"The uncertainties of these averages have been computed by propagating the  uncertainties from each method and tracer."

**Page 15(ish): It does feel a little disjoint to have section 4.1 talking about the various tracer methods, and yet not have any discussion of the ATLAS/SLIMCAT results until you get to the overall inter-comparison discussion in 4.2. Might some of the ATLAS/ SLIMCAT discussion not merit a subsection of its own.**

Sec. 4.1 discusses different tracers that can be used for any of the measurements only methods and the differences we found between them. There are up to six different results for each of the measurement only methods for each year due to the different tracers that can be used. Each of the CTM methods have only one single solution for each year. The discussion of the CTM methods is included in the overall comparison between all methods in Sec. 4.2. There is no section of its own for the passive subtraction methods using CTMs since we found it repetitive to have another section to discuss only the CTM results.

**Page 16 line 7: Here I think youre using "passive subtraction" to only mean the ATLAS/SLMICAT methods, correct? However, in the opening discussion of the manuscript, you have used "passive subtraction" to describe all of your methods (rightly so, as all involve some kind of estimate of passive ozone). Might be better to use a different term here.**

We changed the term used to "passive subtraction method using CTMs" throughout the text.

**Figure 1. Im curious as to where the cluster of black points ("fliers" actually) with O3 around 4.5-5 ppmv in panels a,b,c and d have "gone" in e and f? Are these cases where there were no OCS or CCl3F measurements? Or are they all hiding under the "e)" and**

**"f)" legends (I hope not). Also, in the former cases (a-d) I would expect that they may be contributing significantly to the "uncertainty" in the fit. Might there be something geophysically unusual about them (their ozone abundance clearly implies as much) that would give you a good basis for discounting them? Also, you might want to think about moving the a-f legends to a different corner of the plot to avoid clutter.**

We have used the recommended ACE-FTS quality flags as Sheese et al. (2015) suggested for all species. In the case of OCS and $CClF_3$ no extreme values were left after applying the quality flags. For the other trace gases outliers with high ozone can be seen, however, since we applied the quality flag filter as recommended we have no justification to remove these.

**Figure 2: I'd move the legend (January, March) somewhere else so it doesnt get in the way. Also you dont need it on all four panels (you only had it on one panel in figure 1). That should make it easier for you to find an out of the way place.**

Figure 2 has been changed accordingly.

**Figure 5, caption, line 2: "...2011, with the combined regression fit for January and March...", assuming thats a correct interpretation.**

We have changed the sentence as suggested.

"Panel (a) shows a comparison between the SLIMCAT ozone and ACE-FTS ozone dataset inside the polar vortex for January (black dots) and March (green dots) 2011,  with the combined regression  fit for January and March  shown as a red line. "

---

## Author Comment (AC3) · 9 Jul 2018

We would like to thank Björn-Martin Sinnhuber for his corrections and recommendations. Additions to the text are highlighted in blue and text that has been removed from the original text is highlighted in red. The reviewer comments are included in bold.

**General comments:**
**1) artificial tracer: The argument that mixing does not change the correlation between ozone and the artificial tracer (p4, l4) is only true if the correlation exhibits the same slopes inside and outside the polar vortex. If not, than mixing across the vortex edge can influence the correlation. On p7, l30 it is stated that this method provides a mixing correction. This is not immediately clear. As this is a critical point, I suggest to show the correlations inside and outside of the vortex.**

Mixing of outside polar vortex air affects the correlation of the artificial tracer as air outside the polar vortex does not follow the same correlation. We have included the following statement at the end of the paragraph to clarify this:

"While it reduces the error from mixing of air near the vortex edge, this method, however, does not account for mixing of extra-polar vortex air into the vortex. The artificial tracer, established from observations inside the polar vortex does not follow the same linear correlation outside the polar vortex (Jin et al., 2006)."

**2) Uncertainty of passive subtraction with ATLAS (p11, l8): Estimating the uncertainty by comparing ATLAS and ACE-FTS for January will almost certainly underestimate the true uncertainty, as the model was initialized in early January and run only for a relatively short period uncertainties in model transport will accumulate until March, not captured here. While it is difficult to come up with a better uncertainty estimate, this needs to be at least acknowledged and discussed.**

We have included the following sentence in Sect. 3.4.1 to point out this issue:

"Note that the uncertainty estimated here is a lower bound on the actual uncertainty since it does not consider the accumulated uncertainties in model transport since the initialization in January (e.g. caused by deficiencies in ERA Interim)."

**Specific comments and technical corrections:**

**P3, l13: Polar Night Jet Oscillation Event: I suggest either to give more information or drop the reference to the Polar Night Jet Oscillation Event. What is this and why is this relevant?**

A discussion of the Polar Night Jet Oscillation is probably too specific in the context of this paper. We have changed the sentence to:

"In January 2012, very strong polar vortex disturbance occurred  (Berhard et al., 2012; Chandran et al., 2013) ."

**P4, l13: estimate differences between model and observations: The meaning of this sentence is not fully clear and should be rephrased accordingly.**

This sentence has been removed as it was not relevant.

**P4, l20: ...and the passive subtraction method using only modelled ozone: If the meaning here is ...and compare this to the modelled chemical ozone loss better say so.**

The sentence has been changed accordingly to:

"Chemical ozone depletion for each spring is estimated using the tracer-tracer correlation method, the artificial tracer approach, the average vortex profile descent technique, the modelled passive ozone subtraction method using a Lagrangian and an Eulerian transport model, and modelled chemical ozone loss using SLIMCAT (Chipperfield et al., 2006)."

**P7,l11: high altitudes: upper stratosphere and mesosphere?**

We have changed this sentence accordingly to:

"The tracer-tracer correlation method also neglects descent of ozone or the tracer from high altitudes  (upper stratosphere and mesosphere) above 550 K that is not included in our calculation of the early vortex reference function."

**P13, l11: uncertainty 10-20%: absolute or 10-20% of the ozone loss?**

To clarify, we have changed the sentence to:

"The estimated uncertainties of the ozone loss profile are ∼0.2-0.6 ppmv, or approximately ∼10-20 % of the estimated ozone loss, and the results from all tracers agree within the uncertainties ..."

**P13, l13: that further confirmed the tracer/tracer correlation method to be inaccurate for estimating Arctic ozone loss: This is a strong**

**statement. Do you really want to say tracer/tracer methods are inaccurate for ozone loss estimates?**

Numerous studies (including the references cited here; e.g., Michelsen et al., 1998 a, b, 2000; Plumb et al., 2000, 2003; Plumb, 2007) have shown both theoretically and observationally that trace correlation methods are inaccurate. However, we have softened the language and rephrased the last two sentences of this paragraph:

"This indicates the  shortcomings of the tracer-tracer correlation method, even  in cases where only inner core vortex measurements were used for estimating the ozone loss. These results are  consistent with previous studies (e.g., Michelsen et al., 1998 a, b, 2000; Plumb et al., 2000, 2003; Plumb, 2007) that  have shown tracer-tracer  correlations are not expected to be accurate for estimating Arctic ozone loss. However, in this study, though the profile loss estimates are different for different tracers, the partial column losses (maximum and mean) are not significantly different and agree within the estimated uncertainties. "

**P14, l24: does this apply specifically to ACE-FTS retrievals of OCS and CCl3F ? If so it would be good to mention explicitly.**

We have changed the sentence accordingly to:

'However, using OCS or $CCl_3F$ as a tracer, at least for the ACE-FTS v3.5 dataset, seems to result in larger uncertainties and has the disadvantage that there are not as many profiles available as there are for the rest of the tracers."

**P16, l6: The results of the artificial tracer technique should be uninfluenced by mixing: Again, it needs to be demonstrated that this is also true for mixing across the vortex edge.**

We have changed the paragraph to:

"Discrepancies are apparent between the measurement only methods and the passive subtraction  using CTMs in 2010, especially for the computed mean partial column loss. Each time the vortex splits and the two parts reunite, extra-vortex air is mixed. In 2010 the polar vortex was very disturbed, therefore,  methods that do not account for the mixing of extra-vortex air (the tracer-tracer  method, the profile descent techniques and the artificial tracer technique) are not reliable for that year since an isolated vortex is essential for these methods . The loss estimates in 2010 using the  measurement only techniques do not agree with the passive subtraction

 using CTMs. Generally, we see the largest differences between the passive subtraction method using CTMs and methods that use measurements only for years with strong turbulence and relatively small ozone loss (see Table 1). For example in 2010,  the passive subtraction methods using CTMs are nearly twice as high for the maximum ozone loss and more than three times as high for the mean ozone loss than the methods that use measurements only. This could either be due to mixing processes unaccounted for in the methods using measurements only or the passive subtraction methods using CTMs  variabilities by overestimating passive ozone."

**P16, l9: The passive subtraction methods may smooth out the year-to-year differences and model results in some years may compute some ozone loss even in the absence of ClOx chemistry: Why?? The meaning and basis for this statement is unclear.**

We have changed the paragraph where this comment has been addressed, see comment above (p.16, l. 6).

**P16, l22: ozone loss has also been estimated using only the SLIM-CAT ozone and passive ozone (SLIMCAT only): Again, I believe this is better expressed as modeled ozone loss.**

We have changed the sentence accordingly:

"The ozone loss has also been estimated using only the SLIMCAT  modelled ozone ("SLIMCAT only")."

**P18, l33: passive subtraction methods using either ATLAS or SLIM-CAT seem to have smaller computed uncertainties: As remarked above, I suspect that for these methods the uncertainties here are systematically underestimated.**

The sentence has been changed to:

"While similar ozone losses were computed for all methods in years with an isolated polar vortex, the passive subtraction methods using either ATLAS or SLIMCAT seem to have smaller computed uncertainties. Note that the uncertainty estimated here is a lower bound on the actual uncertainty since it does not consider the accumulated uncertainties in model transport until March."

**P19, l4: and might smooth out the year-to-year variability: again, any idea why the year-to-year variability may be smoothed out?**

The sentence has been changed to:

" 2010,  the passive subtraction methods using CTMs are nearly twice as high for the maximum ozone loss and more than three times as high for the mean ozone loss than the methods that use measurements only. This could either be due to mixing processes unaccounted for in the methods using measurements only or the passive subtraction methods using CTMs may overestimate passive ozone. "

**P19, l11: For years with little to no ClOx activation the artificial tracer correlation technique might be the most reliable because it considers mixing and seems to compute a reasonably small ozone loss: This statement is problematic for two reasons: (a) one may argue that possible mixing across the vortex edge is better represented by the passive subtraction method that takes into account tracer gradients across the vortex edge at least in first order, and (b) the relatively good agreement between the passive subtraction method and modeled ozone loss (SLIMCAT only) for this year (2010) indicates that according to our understanding of the processes involved there was potential for substantial chemical loss.**

We have changed this paragraph to:

"  For years with an unstable polar vortex we recommend using the passive subtraction  technique, since the artificial tracer  technique does not account for mixing of extra-polar vortex air. We did not find any significant difference between an Eulerian or a Lagrangian model and found that both types of CTMs seem to compute the Arctic ozone loss equally well."

---

## Referee Report (RR1)

**Second Review of "Stratospheric ozone loss in the Arctic …"**

BY GRIFFIN ET AL.

**General remarks**

The authors have done extensive revisions of the paper in response to the comments received from all three reviewers. I think that the paper has been improved very much. I think, however, that some of my original concerns have not been fully addressed in the revision (see below). I have also some further comments when reading the new version of the paper again. I apologise that not all of these points were clear to me when writing the first review.

I recommend a further revision of the paper and the improvement of the discussion, as outlined below.

**Discussion**

It is stated in the abstract (and the conclusions) that the average vortex profile descent technique leads to smaller maximum losses. It would be good to quantify this statement somewhat. For example saying something like "by about 15-20 DU"

It is also stated in the abstract that the passive subtraction method using CTMs "results in smaller uncertainties...". However, the reported error bars clearly only cover some of the processes contributing to uncertainties. The CTMs are driven by ERA-I, using a different reanalysis would certainly result in a different vertical transport and thus in different passive ozone profiles. This issue should be discussed in the section on the method (3.4). Further, this method should also be influenced by the 2-5% difference between MLS and ACE-FTS ozone (p. 11). It would also be helpful to estimate the magnitude of this source of uncertainty and report this in section 3.4.

The paper correctly points out that mixing across the vortex edge is an issue for the tracer-tracer method. However, this issue is also implicitly present in the other methods. For example it is not clear how well CTMs represent mixing across the

vortex edge compared to reality. They are very likely not prefect in this respect and therefore will also likely misrepresent to some extent the impact of mixing on passive ozone. I suggest that the impact of mixing across the vortex edge is discussed for all methods used here (section 3).

A focus of the present paper is the chemical Arctic ozone loss in winter 2010/2012. The authors have provided a good coverage of the literature on this case. For this winter, Isaksen, et al. (GRL, 2012) report that "weakened transport of ozone from middle latitudes, concurrent with an anomalously strong polar vortex, was the primary cause of the low ozone. When the zonal winds relaxed in mid-March 2011, Arctic column ozone quickly recovered". The authors might consider to also discuss this facet of polar ozone in winter 2010/2011 in their paper.

In the summary and conclusions it is pointed out that CFC-11 has limited coverage compared to other species and therefore should not be used to estimate ozone depletion. It is not clear however, why this problem does not carry over to the artificial tracer method (discussed just below in the summary). I suggest more discussion of this potential problem.

At the end of the summary and conclusions (p, 20, l. 18) it is stated that the recommendation is to use the passive subtraction technique for years with an unstable polar vortex. This statement (together with table 1) leaves me with the problem that the two CTMs lead to different results for a more stable vortex (e.g. 2005); which model should I believe for 2005? The reason why the passive subtraction technique is recommended (p. 20, l. 18) is that the method took into account mixing across the vortex edge, this should also be okay for 2005, when this mixing was likely less strong.

Finally, I respect that the authors make a different assessment of some of the literature than the reviewer. However, Michelsen et al., (1998b) use an outside vortex reference to discuss the results of the tracer-tracer method; this is clearly not appropriate (and is not done in this paper, see section 3.1). Plumb et al., (2000), in their (very diffusive) conceptual model ignore the fact that there are two very different ozone tracer correlation inside and outside of the vortex edge. These two studies (and the corresponding discussions in review papers) are therefore of limited help when addressing the impact of mixing across the vortex edge on tracer-tracer relations.

**Points remaining from the first review**

I stated in my first review that I suggest improving the discussion of the issue that the tracer-tracer and the artificial tracer methods neglect descent from high altitudes; in particularly from the upper stratosphere and mesosphere. In response, the authors have changed the discussion and pointed out that this effect is not relevant for the height range of interest in the paper (see also their response to my previous comment on p. 10, l. 9).

I agree, but I still think it is necessary to clearly state in the paper that none of the methods discussed here does account for intrusions of mesospheric air. Perhaps I am wrong, but then the mechanism of how descent from high altitudes is included should be given in the description of the methods. Indeed I could only find one paper (Muller et al., JGR, 2007) where the issue of intrusions of mesospheric air into the polar vortex has been discussed in the connection with polar ozone loss estimates

I pointed out in my first review that for the winter 2004/2005, the comparison between the SLIMCAT simulations (SLIMCAT only and SLIMCAT) and the estimated ozone loss from the descent method is and $67 \pm 3$ DU $47 \pm 4$ DU. (Are the authors sure that exactly the same value is correct here for SLIMCAT only and SLIMCAT?) Thus these two methods do not agree for 2005 and 2005 was not a particularly disturbed vortex. However, the two tracer techniques, for 2005 (Table 1) agree within error bars with all methods, both for mean and max (with the exception of Decent, max). Therefore I am confused about the statement in the abstract that "the tracer-tracer correlation method does not agree with other estimation methods in March 2005". This statement seems to be in contradiction to the information given in Table 1.

**Details**

- p. 2., l. 5: not only "include"; these are all the methods. The same is true for the conclusions.

- p. 2., l. 8: "From the seven tracers measured by ACE-FTS...' Perhaps even list the seven tracers.

- p. 2., l. 16: add 'using the different methods" after "loss"

- p. 2., l. 33: change to "primarily ice particles"

- p. 3., l. 1: Here you need to drop "ice" (which does not form at 195-197 K)

- p. 7., l. 21/22: drop "and stable"

- References: Coy et al: title should not be in capital letters

- References: Michelsen et al., 1998 should have a,b in the reference list

- References: Sheese et al., 2016 is listed as "in press"

---

## Referee Report (RR2)

**Final details "Stratospheric ozone loss in the Arctic …"**

BY GRIFFIN ET AL.

**General remarks**

In my opinion the authors have done a very good job in responding to the remaining comments and (subject to a few technical issues) I recommend now acceptance of the paper.

I apologise for perhaps appearing a bit difficult on some issues but I believe that the authors have now presented a very good, balanced study on polar ozone loss estimates; the particular strength of the paper being the application of different methods to the same data set (ACE-FTS).

**Technical comments**

All comments refer to the "track-changes" version of the manuscript.

In the paper there is some discussion on the PSC results from the CALIOP instrument; while I am not pushing a particular citation, I'd like to note that there is a recent publication on the subject (Pitts et al., 2018).

- p 2., l 21: " the analysis of ozone loss"

- p. 3., l. 34: "than *in* in the Antarctic"

- p. 7, l. 27: Mueller et al (2007) is only for one particular year, so I suggest formulating: " In a case study for the year XX, Muller et al. (2007) showed …"

- p. 19., l 25: replace "popular" by "well established"

- References: check consistency with ACP style throughout; some clean-up could be done: page numbers like 'ACH6-1,SOL18-1' should be dropped and replaced by a electronic id (eid in bibtex), p. 29, l. 7: replace 'Ra' by 'Res', p 29. l. 23: update ACPD, p. 30, l. 9: 'Günther', p. 30., l. 26: Fast-Track ?

**References**

Pitts, M. C., Poole, L. R., and Gonzalez, R.: Polar stratospheric cloud climatology based on CALIPSO spaceborne lidar measurements from 2006 to 2017, Atmos. Chem. Phys., 18, 10 881–10 913, doi:10.5194/acp-18-10881-2018, URL `https://www.atmos-chem-phys.net/18/10881/2018/`, 2018.

---

## Author Response (AR2)

We would like to thank the referees for their corrections and recommendations for our revised paper. The reviewer comments are included in bold, with the responses following.

**Referee #1:**

**Minor correction: The new sentence on page 15, line 23 "Note that this does not represent the true uncertainty...is likely much higher" sounds unnecessarily complicated and may be better split in two sentences.**

We have reworded the original sentence.
From:
"Note that this does not represent the true uncertainty but more of a statistical uncertainty, since there is only one passive ozone profile for each March (and, therefore, the same amount of ozone at each potential temperature level) the uncertainty is likely much higher."
To:
"Note that this does not represent the true uncertainty and represents a statistical uncertainty. The true uncertainty is likely much higher, since only one passive ozone profile for each March is applied (and, therefore, the same amount of ozone at each potential temperature level)."

**Referee #3:**

**Overall, I am happy with the changes made to the manuscript but the authors in response to comments from me and the other reviewers. I just have one comment and some minor corrections to suggest.**

**I specifically asked the question about why the MATCH approach was not considered in order that the authors could address it in the text itself, not only in a response to my earlier review. The answer given in the review response was succinct and clear, and I urge the authors to include a sentence in the final manuscript along those lines, if only to deter others from embarking on such an unproductive exercise in the future.**

We have added a couple of sentences in the conclusions addressing the Match approach with ACE-FTS:
"The Match approach is also a popular method used to estimate ozone depletion. However, we found that with the ACE-FTS dataset, it was not possible to estimate the loss between between January and end of March using this method. Due to the orbit of ACE-FTS there is a measurement gap in the Arctic in February that is typically 2-3 weeks, which is too long for the trajectory estimations used for the Match approach."

**Minor comments (Important note: all these are referred to page numbers in the "Tracking changes" [i.e., latexdiff] version of the manuscript attached to the review response).**

**— Page 2**

**Line 3: I suggest "estimation methods based on the same single observation dataset to determinne..."**

We have included the suggestion accordingly.
From:
"For the first time, an evaluation has been performed of six different ozone loss estimation methods with one dataset to determine the Arctic ozone loss (mixing ratio loss profiles and the partial column ozone losses between $380\,K$ and $550\,K$)."
To:
"For the first time, an evaluation has been performed of six different ozone loss estimation methods based on the same single observation dataset to determine the Arctic ozone loss (mixing ratio loss profiles and the partial column ozone losses between $380\,K$ and $550\,K$)."

**Line 9: I suggest ading "also" before "measured"**

This has been added.

**—- Page 3**

**Line 33: "More challenging" than what? The Antarctic clearly, but proper English demans that that be made clear.**

We added the suggestion:
"Because of the strong dynamical variability of the Arctic polar vortex, quantifying chemical ozone loss is more challenging in the Arctic than the Antarctic."

**—- Page 7**

**Line 34: I suggest making "displayed are..." a new sentence.**

We have changed the sentence.
From:
"Figure 1 shows the $O_3$-tracer correlation between for these six tracers for the winter/spring 2011, displayed are the ACE-FTS measurements in January (black dots) and March (green dots) together with the estimated early vortex

reference function (red solid line)."
To:
"Figure 1 shows the $O_3$-tracer correlation between for these six tracers for the winter/spring 2011. The ACE-FTS measurements are displayed in this figure in January (black dots) and March (green dots) together with the estimated early vortex reference function (red solid line)."

—- **Page 15**

**Line 1, "case" -> "cases"**

This has been changed accordingly.

**Referee #2:**

**It is stated in the abstract (and the conclusions) that the average vortex profile descent technique leads to smaller maximum losses. It would be good to quantify this statement somewhat. For example saying something like by about 15-20 DU**

We have changed the sentence accordingly:
From:
"However, the tracer-tracer correlation method does not agree with the other estimation methods in March 2005, and using the average vortex profile descent technique typically leads to smaller maximum losses compared to all other methods."
To:
"However, the tracer-tracer correlation method does not agree with the other estimation methods in March 2005, and using the average vortex profile descent technique typically leads to smaller maximum losses (by approximately 15-30 DU) compared to all other methods."

**It is also stated in the abstract that the passive subtraction method using CTMs results in smaller uncertainties.... However, the reported error bars clearly only cover some of the processes contributing to uncertainties. The CTMs are driven by ERA-I, using a different reanalysis would certainly result in a different vertical transport and thus in different passive ozone profiles. This issue should be discussed in the section on the method (3.4).**

The sentence in the abstract has been revised:
From:
"The passive subtraction method using output from CTMs generally results in smaller uncertainties and slightly larger losses compared to the techniques that use ACE-FTS measurements only."

To:
"The passive subtraction method using output from CTMs generally results in slightly larger losses compared to the techniques that use ACE-FTS measurements only."

We have added the following sentences in Sect. 3.4:
"Both models used in this study are driven by driven by the European Centre for Medium-range Weather Forecasts Reanalysis Interim (ECMWF ERA In terim) meteorological reanalysis (Dee et al., 2011 ).",
and in Sect. 3.4.2:
"Note that the uncertainty estimated here is a lower bound on the actual uncertainty since it does not consider the accumulated uncertainties in model transport (e.g. caused by deficiencies in ERA Interim)."

**Further, this method should also be influenced by the 2-5 % difference between MLS and ACE-FTS ozone (p. 11). It would also be helpful to estimate the magnitude of this source of uncertainty and report this in section 3.4.**

We believe that this is already included in the uncertainty by using the difference between the ACE-FTS dataset and ATLAS for January to estimate an uncertainty of the modelled ozone. It is difficult to separate the total uncertainty into different components.

**The paper correctly points out that mixing across the vortex edge is an issue for the tracer-tracer method. However, this issue is also implicitly present in the other methods. For example it is not clear how well CTMs represent mixing across the vortex edge compared to reality. They are very likely not prefect in this respect and therefore will also likely misrepresent to some extent the impact of mixing on passive ozone. I suggest that the impact of mixing across the vortex edge is discussed for all methods used here (section 3).**

In Sect. 3.3 the issue of mixing extra-polar vortex air for the artificial tracer technique has previously been pointed out:
"While it reduces the error from mixing of air near the vortex edge, this method, however, does not account for mixing of extra-polar vortex air into the vortex. The artificial tracer, established from observations inside the polar vortex does not follow the same linear correlation outside the polar vortex (Jin et al., 2006)."
The following sentences have been added and modified in Sect. 3 to highlight the issues of mixing for all methods:
Sect 3.2 (Average vortex profile descent technique): "It is difficult to estimate the true uncertainty in this case, because of the unknown effect of ozone resulting from mixing processes that are not considered for the average vortex descent technique."
Sect. 3.4:

"The passive subtraction methods using CTMs account for mixing of extra-polar vortex air, however, it is difficult to determine how well these mixing processes are represented within those models."

**A focus of the present paper is the chemical Arctic ozone loss in winter 2010/2012. The authors have provided a good coverage of the literature on this case. For this winter, Isaksen, et al. (GRL, 2012) report that weakened transport of ozone from middle latitudes, concurrent with an anomalously strong polar vortex, was the primary cause of the low ozone. When the zonal winds relaxed in mid-March 2011, Arctic column ozone quickly recovered. The authors might consider to also discuss this facet of polar ozone in winter 2010/2011 in their paper.**

We have included the suggested reference in our paper. We have included the following sentence in the introduction:
"The ozone quickly recovered again once the zonal-wind relaxed and the polar vortex weakened at the end of March (Isaksen et al., 2012) "

**In the summary and conclusions it is pointed out that CFC-11 has limited coverage compared to other species and therefore should not be used to estimate ozone depletion. It is not clear however, why this problem does not carry over to the artificial tracer method (discussed just below in the summary). I suggest more discussion of this potential problem.**

Much of the limited coverage is the lack of positive retrieved concentrations, for CFC-11 the concentration is negative, especially at higher altitudes when concentrations are small and the retrieval oscillates around zero. For the artificial tracer technique negative concentrations did not represent an issue since there are multiple tracers used to estimate a linear correlation. However, for the tracer-tracer correlation negative concentrations were not suitable to use and had to be filtered.

**At the end of the summary and conclusions (p, 20, l. 18) it is stated that the recommendation is to use the passive subtraction technique for years with an unstable polar vortex. This statement (together with table 1) leaves me with the problem that the two CTMs lead to different results for a more stable vortex (e.g. 2005); which model should I believe for 2005? The reason why the passive subtraction technique is recommended (p. 20, l. 18) is that the method took into account mixing across the vortex edge, this should also be okay for 2005, when this mixing was likely less strong.**

The maximum loss is the same for both models for all years, the mean loss in 2005 is slightly different, using SLIMCAT yields to a larger column loss. The

only disagreement between the passive subtraction using ATLAS and SLIM-CAT can be seen for the mean ozone loss in 2005. This was a year where the ozone loss was strong, however, also a lot of mixing occurred throughout the 2004/2005 winter (e.g. Manney et al., 2006). So, we agree none of the models or methods is perfect, however we wanted to include a recommendation and based on this analysis it looks like the passive subtraction using either CTM is the best choice. For the mean loss of 2005, it is hard to say which one is closer to the "real" ozone loss.

**Finally, I respect that the authors make a different assessment of some of the literature than the reviewer. However, Michelsen et al., (1998b) use an outside vortex reference to discuss the results of the tracer-tracer method; this is clearly not appropriate (and is not done in this paper, see section 3.1). Plumb et al., (2000), in their (very diffusive) conceptual model ignore the fact that there are two very different ozone tracer correlation inside and outside of the vortex edge. These two studies (and the corresponding discussions in review papers) are therefore of limited help when addressing the impact of mixing across the vortex edge on tracer-tracer relations.**

During the last revision we have cited the recommended publications to provide a more balanced discussion. We have now removed the reference to Michelsen et al. (1998b). We feel that the paper by Plumb et al. (2000), along side the recommended publications, is important to provide a balanced discussion of the tracer-tracer correlation method.

**Points remaining from the first review**

**I stated in my first review that I suggest improving the discussion of the issue that the tracer-tracer and the artificial tracer methods neglect descent from high altitudes; in particularly from the upper stratosphere and mesosphere. In response, the authors have changed the discussion and pointed out that this effect is not relevant for the height range of interest in the paper (see also their response to my previous comment on p. 10, l. 9).**
**I agree, but I still think it is necessary to clearly state in the paper that none of the methods discussed here does account for intrusions of mesospheric air. Perhaps I am wrong, but then the mechanism of how descent from high altitudes is included should be given in the description of the methods. Indeed I could only find one paper (Muller et al., JGR, 2007) where the issue of intrusions of mesospheric air into the polar vortex has been discussed in the connection with polar ozone loss estimates.**

We have included the suggested reference and added the following sentence to Sect. 3.1:

"Müller et al. (2007) showed that mixing of mesospheric air is likely small and would not lead to an over-estimation of the chemical ozone loss."

**I pointed out in my first review that for the winter 2004/2005, the comparison between the SLIMCAT simulations (SLIMCAT only and SLIMCAT) and the estimated ozone loss from the descent method is and 67-3 DU 47-4 DU. (Are the authors sure that exactly the same value is correct here for SLIMCAT only and SLIMCAT?) Thus these two methods do not agree for 2005 and 2005 was not a particularly disturbed vortex. However, the two tracer techniques, for 2005 (Table 1) agree within error bars with all methods, both for mean and max (with the exception of Decent, max). Therefore I am confused about the statement in the abstract that the tracer-tracer correlation method does not agree with other estimation methods in March 2005. This statement seems to be in contradiction to the information given in Table 1.**

We have changed the sentence in the abstract accordingly.
From:
"However, the tracer-tracer correlation method does not agree with the other estimation methods in March 2005, and using the average vortex profile descent technique typically leads to smaller maximum losses (by approximately 15-30 DU) compared to all other methods."
To:
"However, using the average vortex profile descent technique typically leads to smaller maximum losses (by approximately 15-30 DU) compared to all other methods. "
Note, this is a second change to this sentence (first change was made on first point of Referee #2 review).

**Details**

**p. 2., l. 5: not only include; these are all the methods. The same is true for the conclusions.**

We have changed the sentence in the introduction:
From:
"These methods include the tracer-tracer correlation, the artificial tracer correlation, the average vortex profile descent, and the passive subtraction with model output from both Lagrangian and Eulerian chemical transport models (CTMs)."
To:
"The methods used are the tracer-tracer correlation, the artificial tracer correlation, the average vortex profile descent, and the passive subtraction with model output from both Lagrangian and Eulerian chemical transport models (CTMs)."

We have changed the sentence in the conclusions:
From:
"These estimation methods include tracer-tracer correlation, ..."
To:
"The estimation methods used are the tracer-tracer correlation, ..."

**p. 2., l. 8: From the seven tracers measured by ACE-FTS... Perhaps even list the seven tracers.**

We have changed the sentence accordingly.
From:
"From these seven tracers investigated, we found that $CH_4$, $N_2O$, HF, and CFC-12 are the most suitable tracers for investigating polar stratospheric ozone depletion with ACE-FTS."
To:
"From these seven tracers investigated ($CH_4$, $N_2O$, HF,OCS, CFC-11, CFC-12, CFC-113), we found that $CH_4$, $N_2O$, HF, and CFC-12 are the most suitable tracers for investigating polar stratospheric ozone depletion with ACE-FTS."

**p. 2., l. 16: add using the different methods after loss**

We have changed the sentence accordingly:
From:
"The estimated partial column ozone loss inside the polar vortex (between $380\,K$ and $550\,K$) is 66-103 DU, 61-95 DU, 59-96 DU, 41-89 DU, and 85-122 DU for March 2005, 2007, 2008, 2010, and 2011, respectively."
To:
"The estimated partial column ozone loss inside the polar vortex (between $380\,K$ and $550\,K$) using the different methods is 66-103 DU, 61-95 DU, 59-96 DU, 41-89 DU, and 85-122 DU for March 2005, 2007, 2008, 2010, and 2011, respectively."

**p. 2., l. 33: change to primarily ice particles**

This has been changed.

**p. 3., l. 1: Here you need to drop ice (which does not form at 195-197 K)**

This has been changed.

**p. 7., l. 21/22: drop and stable**

This has been changed.

**References: Coy et al: title should not be in capital letters**

This has been changed.

**References: Michelsen et al., 1998 should have a,b in the reference list**

The reference to Michelsen et al., 1998b has been removed as part of the previous suggestion by Referee #2.

**References: Sheese et al., 2016 is listed as in press**

This has been changed.

**We have also corrected the following typos in the manuscript:**

p.10, l.5: For consistency we have changed $R^2$ to $R$; "...$R = 0.9$, the other tracers have $R \geq 0.95$..."

p. 10, l. 17: from: "...dynamic is...." to: "...dynamics are..."

p. 12, l. 27: from: "..geo-location..." to: "...geo-locations..."

p.12, l. 34: from: "...passive and ozone..." to: "...passive ozone and ozone..."

p. 17, l. 12: from: "...profile decent techniques..." to: "...profile descent technique..."

p.20, l. 18: from: "...until March). ..." to: "...until March. ..."

p. 20, l. 21: from: "...vortex, e.g. the passive..." to: "...vortex, the passive..."

p. 35 (caption of Fig. 3): from: "Artificial correlation ..." to: "Artificial tracer correlation..."

[revised manuscript text omitted]

---

## Author Response (AR3)

We would like to thank the referee and the editor for their corrections and recommendations for our revised paper. The reviewer comments are included in bold, with the responses following.

**From the editor:**

**Page 4, line 23: "single dataset"**

From: "single observational dataset"
To: "single dataset"

**Page 5, line 33: "to remove unrealistic outliers ... "**

From: "to remove physically unrealistic outliers"
To: "to remove unrealistic outliers"

**Page 19, line 16: How much is this (0.5 ppmv) in %?**

From: "(with maximum differences of 0.5 ppmv)"
To: "(with maximum differences of 0.5 ppmv, approximately 20 % of the estimated losses)"

**Page 19, line 18: estimated for six Arctic winters? or nine?**

From: "from various methods over a nine year period between the winters of 2004/2005 and 2012/2013"
To: "from various methods for five years between the winters of 2004/2005 and 2012/2013"

**Page 20, line 32: Thank you for the recommendation. However, we still have to be careful about the technique when we estimate the loss in warm winters, as the tracer/ozone values might show a jump at the time of warming period and that can corrupt the loss estimated using the measurements. This may not be the case for all models though.**

Thank you for the comment. As we noted in the summary and conclusions, our recommendations are based on our results. We have adjusted page 20, line 33 as follows to reinforce this.
From: "For years with an unstable polar vortex..."
To: "Based on the years studied, for years with an unstable polar vortex..."

**From the referee report:**

**General remarks**

In my opinion the authors have done a very good job in responding to the remaining comments and (subject to a few technical issues) I recommend now acceptance of the paper. I apologise for perhaps appearing a bit difficult on some issues but I believe that the authors have now presented a very good, balanced study on polar ozone loss estimates; the particular strength of the paper being the application of different methods to the same data set (ACE-FTS).

**Technical comments**

**All comments refer to the "track-changes" version of the manuscript. In the paper there is some discussion on the PSC results from the CALIOP instrument; while I am not pushing a particular citation, I'd like to note that there is a recent publication on the subject (Pitts et al., 2018).**

Thank you for pointing us to the study by Pitts et al., 2018. We have chosen not to include it.

**p 2., l. 21: " the analysis of ozone loss"**

From: "for the analysis."
To: "for the analysis of ozone loss."

**p. 3., l. 34: "than in in the Antarctic"**

From: "than the Antarctic"
To: "than in the Antarctic"

**p. 7, l. 27: Mueller et al (2007) is only for one particular year, so I suggest formulating: "In a case study for the year XX, Muller et al. (2007) showed . . . "**

From: " Müller et al. (2007) showed"
To: " In a case study for the spring 2003, Müller et al. (2007) showed"

**p. 19., l 25: replace "popular" by "well established"**

From: "The Match approach is also a popular method"
To: "The Match approach is also a well-established method"

**References: check consistency with ACP style throughout; some cleanup could be done: page numbers like 'ACH6-1,SOL18-1' should be dropped and replaced by a electronic id (eid in bibtex), p. 29, l. 7: replace 'Ra' by 'Res', p 29. l. 23: update ACPD, p. 30, l. 9: "Günther", p.**

**30., l. 26: Fast-Track ?**

The recommended changes have been made regarding the page number, the update from ACPD to ACP, and the name "Günther". The abbreviation on p.29, l.7 follows the convention of the Caltech Library website and the Fast-Track is from the journal formatting for Annals of Geophysics.

[revised manuscript text omitted]